# A pre-screening strategy to assess resected tumor margins by imaging cytoplasmic viscosity and hypoxia

Hui Huang[1,2], Youpei Lin[3], Wenrui Ma[4], Jiannan Liu[5], Jing Han[5], Xiaoyi Hu[6], Meilin Tang[1], Shiqiang Yan[1,2], Mieradilijiang Abudupataer[4], Chenping Zhang[5], Qiang Gao[3], Weijia Zhang[1,2,4]*

[1]Shanghai Fifth People's Hospital and Institutes of Biomedical Sciences, Fudan University, Shanghai, China; [2]The State Key Laboratory of Molecular Engineering of Polymers and The Shanghai Key Laboratory of Medical Imaging Computing and Computer Assisted Intervention, Fudan University, Shanghai, China; [3]Department of Liver Surgery and Transplantation, Liver Cancer Institute, Zhongshan Hospital, Fudan University, Shanghai, China; [4]Department of Cardiac Surgery and Shanghai Institute of Cardiovascular Diseases, Zhongshan Hospital, Fudan University, Shanghai, China; [5]Department of Oromaxillofacial Head and Neck Oncology, Shanghai Ninth People's Hospital, Shanghai Jiao Tong University School of Medicine, Shanghai, China; [6]Department of Urology, Zhongshan Hospital, Fudan University, Shanghai, China

*For correspondence:
weijiazhang@fudan.edu.cn

Competing interest: The authors declare that no competing interests exist.

**Abstract** To assure complete tumor removal, frozen section analysis is the most common procedure for intraoperative pathological assessment of resected tumor margins. However, during one operation, multiple biopsies may be sent for examination, but only few of them are made into cryosections because of the complex preparation protocols and time-consuming pathological analysis, which potentially increases the risk of overlooking tumor involvement. Here, we propose a fluorescence-based pre-screening strategy that allows high-throughput, convenient, and fast gross assessment of resected tumor margins. A dual-activatable cationic fluorescent molecular rotor was developed to specifically illuminate live tumor cells' cytoplasm by emitting two different fluorescence signals in response to elevations in hypoxia-induced nitroreductase (a biochemical marker) and cytoplasmic viscosity (a biophysical marker), two characteristics of cancer cells. The ability of the fluorescent molecular rotor in detecting tumor cells was evaluated in mouse and human specimens of multiple tissues by comparing with hematoxylin and eosin staining. Importantly, the fluorescent molecular rotor achieved 100 % specificity in discriminating lung and liver cancers from normal tissue, allowing pre-screening of the tumor-free surgical margins and promoting clinical decision. Altogether, this type of fluorescent molecular rotor and the proposed strategy may serve as a new option to facilitate intraoperative assessment of resected tumor margins.

## Introduction

During a tumor resection operation, a pathologist examines the removed tissues to determine whether the resected margins are clear of tumor involvement. The results should be immediately returned to the operating surgeon to guide further treatment such as extended resection (*Lam et al., 2008*; *Marszalek et al., 2012*; *Hinni et al., 2013*). Over the past 30 years, the strategy of frozen section and microscopic examination has been the first-line intraoperative assessment of tumor margins (*Cendán et al., 2005*; *Algaba et al., 2005*). Despite its high specificity, frozen section-based pathological assessment is limited by inadequate sampling and complex protocols, which precludes

comprehensive analysis of all biopsies (*Olson et al., 2011*; *Black et al., 2006*; *St John et al., 2017*). Improvement of the current method or alternative options are required to provide an overall view of the resected tissue with simple and fast preparation.

Recently, fluorescence-guided imaging has emerged as a promising tool in cancer surgery, owing to its simple implementation, excellent selectivity and favorable biocompatibility (*Vahrmeijer et al., 2013*; *Bu et al., 2014*; *Hussain and Nguyen, 2014*; *Hernot et al., 2019*; *Low et al., 2018*). Targeting the tumor markers on the cell membrane, fluorescent probes have also been developed to assess the resected surgical margins (*Hoogstins et al., 2016*; *Gao et al., 2018*; *van Keulen et al., 2019*; *Koller et al., 2018*; *Voskuil et al., 2020*). On the other hand, abnormal intracellular microenvironment has been characterized as a universal hallmark of cancer cells (*Gillies and Gatenby, 2007*; *Yang et al., 2014*; *Hui and Chen, 2015*; *Catalano et al., 2013*). Hypoxia, which is partly caused by exaggerated growth of tumor cells, can induce the overexpressions of biochemical endogenous reductases such as nitroreductase, azoreductase, quinone reductase, and others (*Brown and Wilson, 2004*). From the biophysical viewpoint, a relatively higher cytoplasmic viscosity is another distinctive feature of cancer cells (*Yang et al., 2014*; *Kalwarczyk et al., 2011*; *Claus, 1942*), where changes in viscosity of the mitochondrial matrix are reported to correlate with abnormality of mitochondrial respiration and metabolism (*Singer and Nicolson, 1972*; *Kuimova et al., 2009*; *Lee et al., 2021*). To date, different types of fluorescent probes have been developed to visualize either intracellular hypoxia (*Liu et al., 2017*; *Zhao et al., 2020*) or change in cytoplasmic viscosity (*Kuimova et al., 2009*; *Hao et al., 2019*, *Ye et al., 2021*; *Wolstenholme et al., 2020*; *Zhou et al., 2021*), which might be susceptible to fluctuations in the intracellular microenvironment. Given this, we postulate that an integrated evaluation of the biochemical (hypoxia) and biophysical (viscosity) features of tumor cells might enhance the accuracy of fluorescence-based assessment of surgical specimens.

Here, we reported the design, synthesis, and application of a dual-activatable fluorescent molecular rotor, IBS440. It consists of one pyridium derivative and one dimethylaniline group, both of which can intramolecularly rotate around the conjugate domain and cause non-radiative decay that quenches fluorescence emission. A higher viscosity of solution can restrict this intramolecular rotation and emit strong fluorescence signals in the 580–640 nm wavelength range. Meanwhile, the nitro group of IBS440 can be reduced in the presence of nitroreductase, a hypoxia-induced endogenous enzyme, thereby triggering elimination reaction to form a new compound termed IBS224 with fluorescence emission at 500–550 nm. Further, we evaluated the activability of this fluorescent molecular rotor in liquid solutions, live cells and tumor-bearing mice tissues in vivo and ex vivo. Tumor involvement in surgical specimens of four cancer types was assessed and compared between fluorescence and hematoxylin and eosin (H&E) staining of permanent sections. Finally, exploiting the 100 % specificity of this molecule in screening tumor-free tissues, we proposed a fluorescence-based strategy to pre-screen biopsies in a rapid, high-throughput manner, which may facilitate clinical decision.

## Results

### Design and synthesis of IBS440

First, we developed a dual-activatable fluorescent molecular rotor IBS440 to monitor the increased cytoplasmic viscosity and hypoxia-induced nitroreductase. Based on the principle of twisting intramolecular charge transfer, we designed a molecular rotor to monitor viscosity. The rotor showed a weak fluorescence intensity in low-viscosity situation, which was ascribed to the free intramolecular rotation between pyridium derivative and dimethylaniline. With increasing viscosity, the intramolecular rotation was restricted, releasing a strong red fluorescence. Meanwhile, nitrobenzene was introduced into the chemical structure as an alternative to nitroreductase substrate, which interacted with nitroreductase under the existence of nicotinamide adenine dinucleotide phosphate (NADPH) as electron donor; and then the nitro group of IBS440 was reduced to amine group triggering elimination reaction to obtain a new compound, named IBS224. Due to the mechanism of intramolecular charge transfer, a strong green fluorescence signal was emitted from IBS224. Chemical structures of IBS440 and IBS224 were characterized by [1]H NMR, [13]C NMR, and HRMS spectra (see Appendix 1: the [1] H NMR, [13] C NMR, and HRMS spectra.).

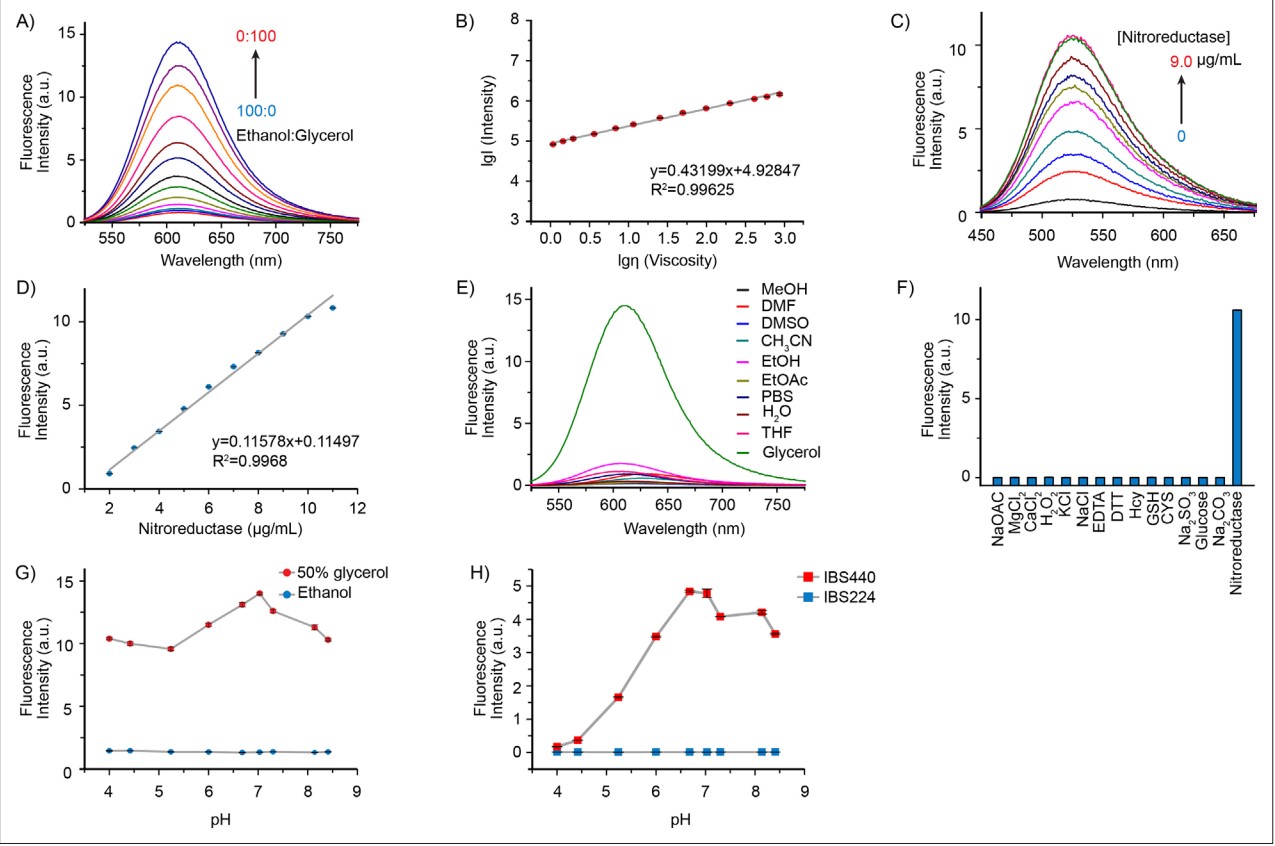

**Figure 1.** Fluorescence emission and absorption profiles of IBS440. (**A**) Fluorescence spectra of IBS440 (10 µM) in different ratios of Ethanol/Glycerol mixtures. (**B**) The linear response between the fluorescence intensity at 610 nm (lgl(Intensity)) of the probe IBS440 (10 µM) and the viscosity (lg $\eta$ (Viscosity)) in the Ethanol/Glycerol solvent. (**C**) The fluorescence response of IBS440 (10 µM) to nitroreductase at the varied concentrations in reaction buffer (50 mM Tris-HCl, 150 mM NaCl, 1 mM DTT, 0.1 mM EDTA, pH 8.0). The spectra were recorded upon treatment of IBS440 (10 µM) with nitroreductase (0–9.0 µg/mL) in the presence of NADPH (500 µM). (**D**) A linear correlation between the concentration of nitroreductase and the fluorescence intensity of the reaction mixture. (**E**) Fluorescence intensity of IBS440 (10 µM) at 610 nm in various solvents of methanol (MeOH), N, N-Dimethylformamide (DMF), dimethyl sulfoxide (DMSO), acetonitrile (CH$_3$CN), ethanol (EtOH), ethyl acetate (EtOAc), phosphate buffer saline (PBS), H$_2$O, tetrahydrofuran (THF), glycerol. (**F**) Fluorescence responses of IBS440 (10 µM) in the presence of NADPH (500 µM) to various species. $\lambda_{ex}$ = 400 nm. $\lambda_{em}$ = 520 nm. (**G**) Effects of pH on the response of IBS440 in solvents with different viscosity (ethanol and 50 % glycerol in ethanol). The fluorescence intensity at 610 nm was plotted against different pH values. $\lambda_{ex}$ = 500 nm. (**H**) The emission intensity (at 520 nm) of IBS224 and IBS440 at different pH Tris-HCl buffer, containing 20 % DMSO as a cosolvent. Error bars represent standard deviation of three repeated experiments.

The online version of this article includes the following figure supplement(s) for figure 1:

**Source data 1.** The original raw data of fluorescence emission and absorption profiles.

**Figure supplement 1.** Confirmation and immunoblotting analysis of nitroreductase.

**Figure supplement 2.** Absorption spectra of the probe IBS440 (10 µM), IBS440 (10 µM)+ NADPH (500 µM)+ nitroreductase (9 µg/mL), IBS224 (10 µM) in reaction buffer, containing 20 % DMSO as a co-solvent.

**Figure supplement 3.** Photostability test of IBS440 (10 µM) in glycerol for 3600 s.

## Fluorescence emission and absorption profiles of IBS440

The fluorescence and Ultraviolet-visible (UV-vis) absorption spectrums of IBS440 were measured to evaluate its photophysical features in response to viscosity and nitroreductase. Recombinant *E. coli* nitroreductase was expressed and purified as a fusion protein with a His tag. The protein molecular weight was 27 kD and its purity were confirmed by sodium dodecyl sulfate polyacrylamide gel electrophoresis (SDS-PAGE) (*Figure 1—figure supplement 1*). The fluorescence intensity of IBS440 in different ratios of ethanol and glycerol mixtures was investigated. With an increase in the proportion of glycerol, accompanied with an increased viscosity, the corresponding emission intensity of IBS440 was significantly enhanced compared with in pure ethanol (*Figure 1A*). There was an approximately 18-fold enhancement of intensity from ethanol to glycerol. A good linear relationship existed between

fluorescent intensity (lgI) and viscosity (lg $\eta$ ), with a correlation coefficient of 0.994 (*Figure 1B*), indicating it enabled quantitative determination of viscosity. To test the recognition of nitroreductase, the spectral response of IBS440 to nitroreductase was assessed in reaction buffer with 500 µM NADPH as cofactor (*Figure 1C*). Upon the addition of nitroreductase (0–9.0 µg/mL), the nitro group of IBS440 was reduced to amine group, thereby triggering an elimination reaction to produce a new compound IBS224. The blue-shifts in wavelength were observed in the UV-vis absorption spectra of IBS440 and IBS224 (*Figure 1—figure supplement 2*). Meanwhile, the fluorescence intensity of IBS440 was found to be linearly proportional to nitroreductase concentration within the range 0–9.0 µg/mL (*Figure 1C and D*). The limit of detection (LOD) was calculated using the equation of LOD = 3σ/k, wherein σ was the standard deviation, and k was the slope of the linear line. According to the linear correlation, k was determined to be 11166.424, and σ was set to 77.883, leading to a sensitive detection limit of 20.9 ng/mL (*Figure 1D*). It suggested that this molecule can detect nitroreductase sensitively in liquid solutions.

To confirm whether the fluorescence of IBS440 was influenced by polarity changes, the fluorescence spectral profiles of IBS440 in various polarity solvents were evaluated. As shown in *Figure 1E*, IBS440 exhibited a strong fluorescence emission in glycerol, in contrast, it showed much weaker fluorescence in other solvents. To examine the nitroreductase selectivity of IBS440, its fluorescence intensity was measured upon addition of various biologically relevant cations. Compared to some ions ($Na^+$, $Mg^{2+}$, $Ca^{2+}$, $K^+$, $Cl^-$, $AcO^-$, $CO_3^{2-}$, $SO_3^{2-}$) and biologically relevant species (Cys, Hcy, GSH, $H_2O_2$, EDTA, DTT, glucose), IBS440 showed a specific fluorescence emission in nitroreductase in the presence of NADPH (*Figure 1F*). These results illustrated that IBS440 was insensitive to the polarity of the solvents and highly selective to nitroreductase, suggesting it can be used to detect viscosity and nitroreductase changes in complex microenvironments.

In addition, as the pH across organelles varies, it is important that the fluorescence feature of IBS440 is not affected by pH variation. We measured the effects of pH (4.0–9.0) on the fluorescence of IBS440 in pure ethanol and 50 % glycerol-ethanol mixture as well as the fluorescence of IBS440 and IBS224 in reaction buffer. The emission values of IBS440 at 610 nm in ethanol and at 520 nm in reaction buffer had almost no change, while weak change of IBS440 in 50 % glycerol media over the range of pH 4.0–9.0 and optimum value of IBS224 in pH 7.0 was observed (*Figure 1G and H*), demonstrating that IBS440 was stable at the physiological pH conditions and may promote its application in biological systems. Notably, the fluorescence intensity at 610 nm was decreased about 0.6 % when IBS440 was exposed to 500 nm excitation laser for 10 min and decreased 2 % after 1 hr, showing good photostability of IBS440 (*Figure 1—figure supplement 3*). Therefore, these data exhibited the sensitivity and specificity of IBS440 respond to viscosity and hypoxia-induced nitroreductase in liquid solutions.

## Fluorescence localization of IBS440 in live cells and ex vivo tumor tissues

To explore the feasibility of applying IBS440 into live cell system, we first evaluated its cytotoxicity by a standard CCK8 assay in three different cell lines including MHCC97H, A549 and FaDu cells (*Figure 2—figure supplement 1*). We incubated the cells with IBS440 at a gradient concentration from 5 µM to 50 µM for 12 hr or 24 hr. The cell viabilities were higher than 70 % even the concentration was increased to 50 µM and the incubation time was 24 hr. The following experimental concentration we chose was 5 µM considering that its cell viability was above 90 %. We then tested the subcellular localization of IBS440. Cells were co-stained with a commercially available green fluorescent mitochondrial dye (Mito-Tracker Green) and IBS440. The merged image of red channel was well overlaid with the green channel leading to colocalization coefficient (Pearson's correlation) of 0.9363 for MHCC97H cells, 0.9386 for FaDu cells and 0.9257 for A549 cells (*Figure 2A*), implying that IBS440 mainly localized in mitochondria of live cells. To further measure the ability of IBS440 to illuminate tumor tissues, we incubated the fresh mice tumor slices with IBS440 and imaged on microscopic level. Freshly excised tissue represents the best in vivo situation for the determination of localization. Hoechst 33,342 is a specific dye for AT-rich regions of double-stranded DNA and suitable for DNA, chromosome and nuclear staining. We found that the red fluorescence of the molecule mainly occurred around the cell nuclear (blue fluorescence) in tumor slices (*Figure 2B*), indicating IBS440 enabled to light up tumor cells in fresh tumor tissues.

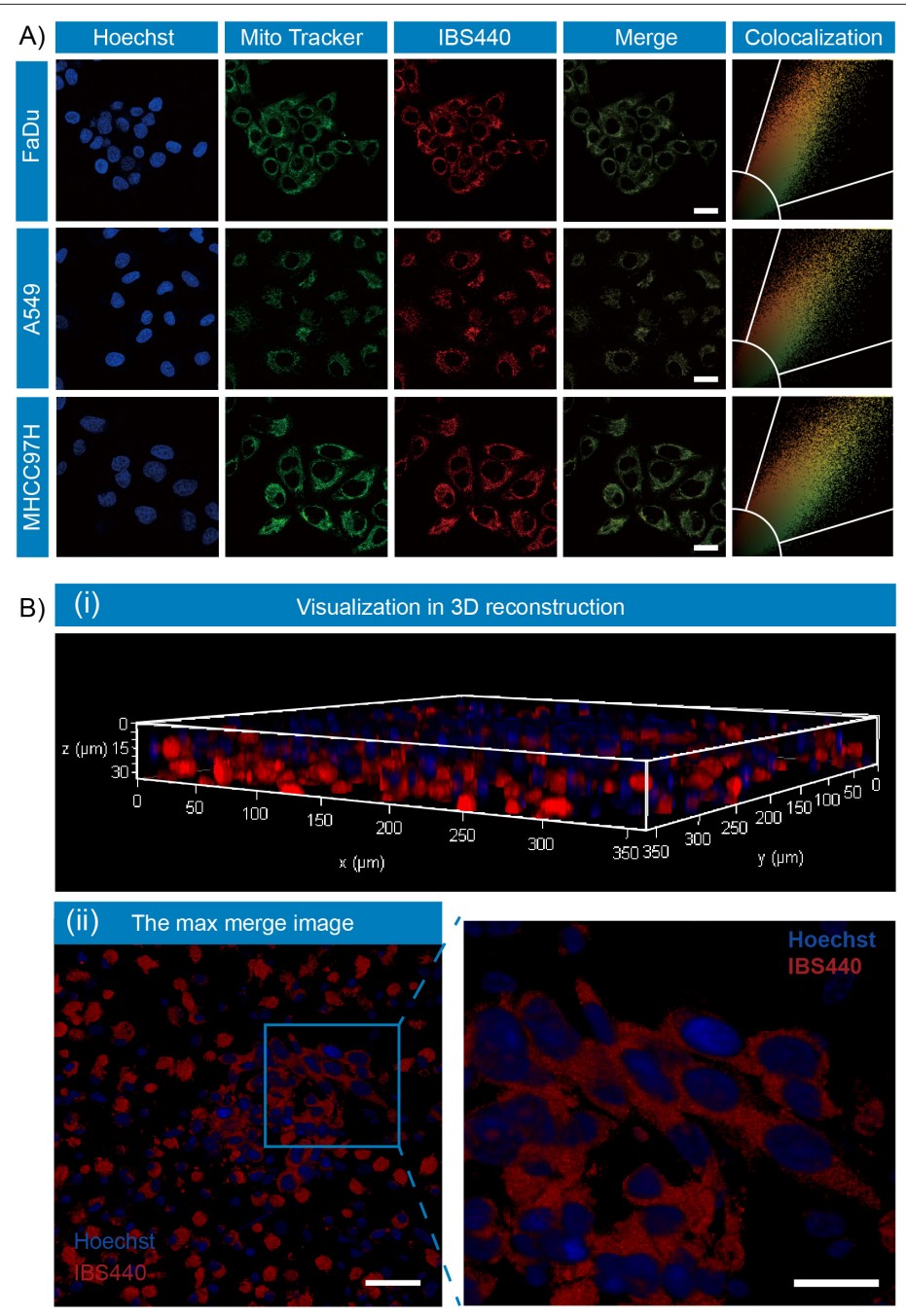

**Figure 2.** Fluorescence localization of IBS440 in live cells and ex vivo tumor tissues. (**A**) The cells were treated with IBS440 (5 μM) for 20 min and then stained with Hoechst 33,342 (1x) and Mito Tracker Green (10 μM) for 10 min. Blue channel ($\lambda_{ex}$ = 405 nm, $\lambda_{em}$ = 430–480 nm). Green channel ($\lambda_{ex}$ = 488 nm, $\lambda_{em}$ = 500–550 nm). Red channel ($\lambda_{ex}$ = 488 nm, $\lambda_{em}$ = 580–630 nm). Scale bar: 25 μm. (**B**) Visualization in 3D reconstruction (**i**) and the max merge image of 3D structure (ii). Scale bar: 100 μm.

The online version of this article includes the following figure supplement(s) for figure 2:

**Source data 1.** The original raw data of fluorescence localization.

**Figure supplement 1.** Cell viability test (%) in three cell lines (FaDu cell, MHCC97H cell and A549 cell) incubated with IBS440 or compound IBS224 (5–50 μM) for different incubation time.

**Figure supplement 2.** Fluorescence imaging of cancerous cells (MHCC97H, FaDu and A549 cells) and noncancerous cells (RAW264.7 and A7r5 cells).

*Figure 2 continued on next page*

*Figure 2 continued*

**Figure supplement 2—source data 1.** The original raw data of fluorescence imaging of Mito-Tracker Green.

**Figure supplement 3.** Fluorescence imaging of hepatocellular tumor and normal tissues.

**Figure supplement 4.** HIF-1α expression level in three different cell lines under normoxia (20 % $O_2$) and hypoxia (2 % $O_2$) conditions for 12 hr or 24 hr by immunoblotting.

**Figure supplement 5.** Fluorescence imaging of hypoxia in live cells.

**Figure supplement 5—source data 1.** The original raw data of fluorescence imaging of hypoxia in live cells.

**Figure supplement 6.** Fluorescence imaging of viscosity in cancer cells (MHCC97H, FaDu and A549 cells) and normal cells (RAW264.7 and A7r5 cells) treated with IBS440 (5 µM) for 20 min.

**Figure supplement 6—source data 1.** The original raw data of fluorescence imaging of viscosity in live cells.

**Figure supplement 7.** Visualization of the normal tissues of mice (A).

**Figure supplement 7—source data 1.** The original raw data of fluorescence 3D reconstruction image in tissues.

## Fluorescence imaging of Mito-Tracker Green in live cells and tissues

Fluorescent small molecules are effectively used in imaging fields because of simple synthesis, high sensitivity and noninvasiveness. Because of their lipophilicity and small molecular weight, the way that they enter cells mainly depends on passive diffusion, which is one of the most common and important transportation methods in lipophilic molecule transportation (*Bressloff and Newby, 2013*). As shown in *Figure 2—figure supplement 2*, the cancerous cells (MHCC97H, A549 and FaDu) and noncancerous cells (RAW264.7, A7r5) were incubated with a commercial lipophilic mitochondrial dye Mito-Tracker Green, the result shows the fluorescence intensity in these cells have no significant difference. As shown in *Figure 2—figure supplement 3*, hepatocellular cancer and normal tissues were incubated with Mito-Tracker Green, and there was also no significant difference in fluorescence intensity. These results showed that the uptake of lipophilic small molecules by cancerous cells and noncancerous cells is at the very similar level.

## Fluorescence imaging of hypoxia in live cells

We next validated the fluorescence responses of IBS440 to elevations in nitroreductase in cultured cell lines. Hypoxia inducible factor 1α (HIF-1α) is overexpressed in common human cancers (*Liao et al., 2007*). It was stably expressed when cells were cultured under 2 % $O_2$ for 12 hr or 24 hr, indicating that the cells were under a hypoxic station (*Figure 2—figure supplement 4*). As shown in *Figure 2—figure supplement 5*, fluorescence signal was faint under normoxic condition (20 % $O_2$), suggesting that the level of nitroreductase expressed in cells was low under normoxic condition. The fluorescence intensity increased when the concentration of $O_2$ decreased to 2%, which implied the up regulation of nitroreductase expression. Thus, the hypoxia condition of cells is well reflected by the fluorescence-based detection of nitroreductase. These results evidenced that IBS440 served as a hypoxia monitor in live cells.

## Fluorescence imaging of viscosity in live cells

We further studied the fluorescence of IBS440 in viscosity detection channel of cancerous and noncancerous cells. MHCC97H, A549, FaDu, RAW264.7 and A7r5 cells were chosen for fluorescence imaging by confocal microscopy. As shown in *Figure 2—figure supplement 6*, the cancerous cells (MHCC97H, A549 and FaDu) and noncancerous cells (RAW264.7, A7r5) were stained with IBS440, the fluorescence intensity of the cancerous cells was stronger than that of the noncancerous cells. Specifically, the normal tissues of mice (including heart, liver, spleen, lung, kidney, muscle, fat, and brain) and MHCC97H tumor tissue were treated with IBS440, respectively. In comparison with the dim fluorescence signals from normal tissues, the tumor tissue exhibited significantly enhanced fluorescence (*Figure 2—figure supplement 7*), implying the increased viscosity of tumor tissue.

## In vivo and ex vivo fluorescence imaging of mouse tumor

Next, we evaluated its capability to monitor viscosity and nitroreductase in vivo. We used BALB/c nude mouse models bearing subcutaneous MHCC97H cells, A549 cells and FaDu cells, respectively. Significant signals from both viscosity and nitroreductase fluorescence detection channels were

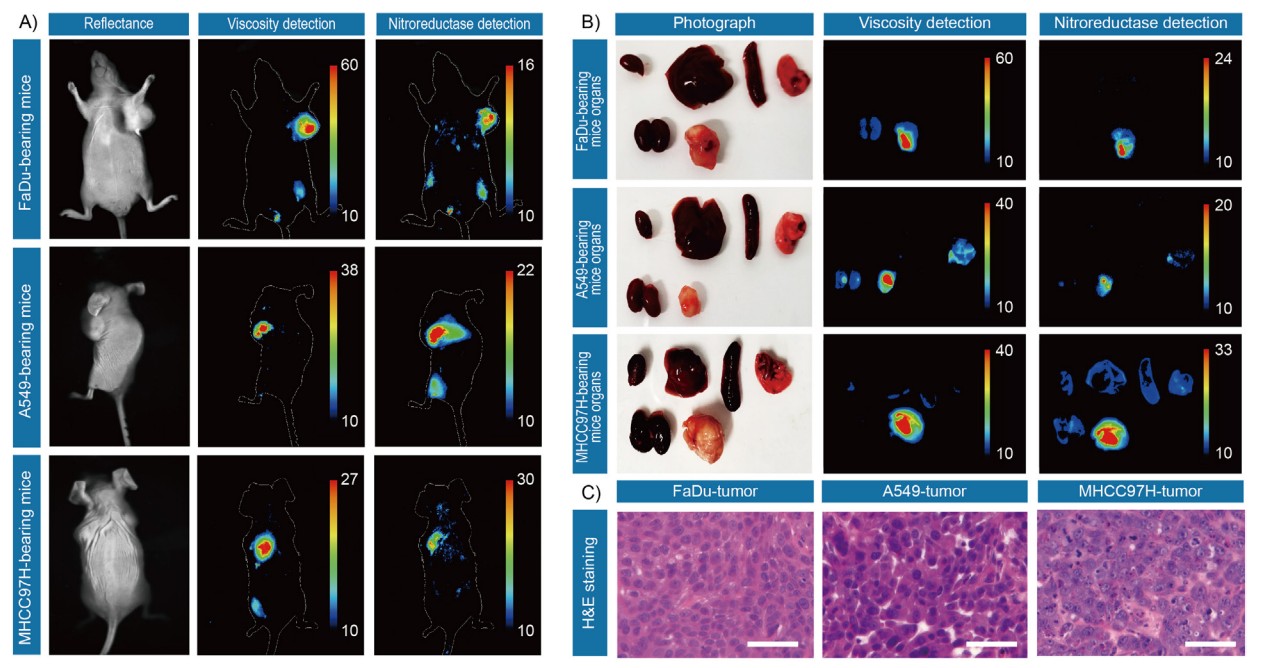

**Figure 3.** In vivo and ex vivo fluorescence imaging of mouse tumors. (**A**) In vivo fluorescence imaging of the tumor-bearing mice after stained with IBS440. (**B**) Ex vivo fluorescence imaging of the major organs (heart, liver, spleen, lung, and kidney) and tumor tissues of tumor-bearing mice. (**C**) Hematoxylin and eosin microscopic imaging of the resected tumor tissues. Scale bar: 100 μm.

The online version of this article includes the following figure supplement(s) for figure 3:

**Figure supplement 1.** Hematoxylin and eosin staining of the major organ of the tumor-bearing mice.

**Source data 1.** The original raw data of fluorescence imaging of mouse tumor.

**Source data 2.** The original raw data of fluorescence imaging of mouse tumor.

observed in the tumor regions, in contrast to the background signals in muscle (*Figure 3A*). To extend the application of IBS440, the major organs (heart, liver, spleen, lung, kidney and muscle) and the tumor tissues of the mice were harvested and incubated with IBS440 for ex vivo imaging. It could be observed that the fluorescence signals of the tumor tissues were much stronger than the major organs (*Figure 3B*). Hematoxylin and eosin staining analysis validated the tumor tissues, confirming that the fluorescence signals were tumor specific (*Figure 3C* and *Figure 3—figure supplement 1*). These results showed that IBS440 responded to viscosity and nitroreductase in tumor-bearing mice in vivo as well as resected tissues ex vivo.

## Ex vivo imaging of patients' resected tumor specimens

The primary aim of our study was to provide assessments of tumor-involved margins in surgical specimens, thus we measured two types of patients' resected specimens including hepatocellular cancer and lung cancer. First, nine pairs of liver cancerous/noncancerous tissue tissues derived from nine individual patients and five pairs of lung cancerous/noncancerous tissues derived from five individual patients (pathologically validated by the subsequent hematoxylin and eosin staining) were used as a training dataset and incubated with IBS440 and imaged by an in vivo small animal instrument, respectively. As shown in *Figures 4 and 5*, *Figure 5—figure supplement 2* and *Figure 5—figure supplement 3*, the stronger fluorescence signals emitted from cancerous tissues and the weaker signals from noncancerous tissues. The maximal gray-scales signal intensities of the same cancer type samples were used to plot a scatter graph. The centroid of scatter points, computed by K-means clustering algorithm, were determined as the threshold. As shown in *Figure 4—figure supplement 1*, *Figure 5—figure supplement 1*, *Figure 5—figure supplement 4* and *Figure 5—figure supplement 5*, the threshold value was calculated as (160.3,164.3) of hepatocellular cancer, (164.4,165.8) of lung cancer, (164.7,165.3) of oral cancer and (163.5, 167.1) of renal cancer. According to the threshold

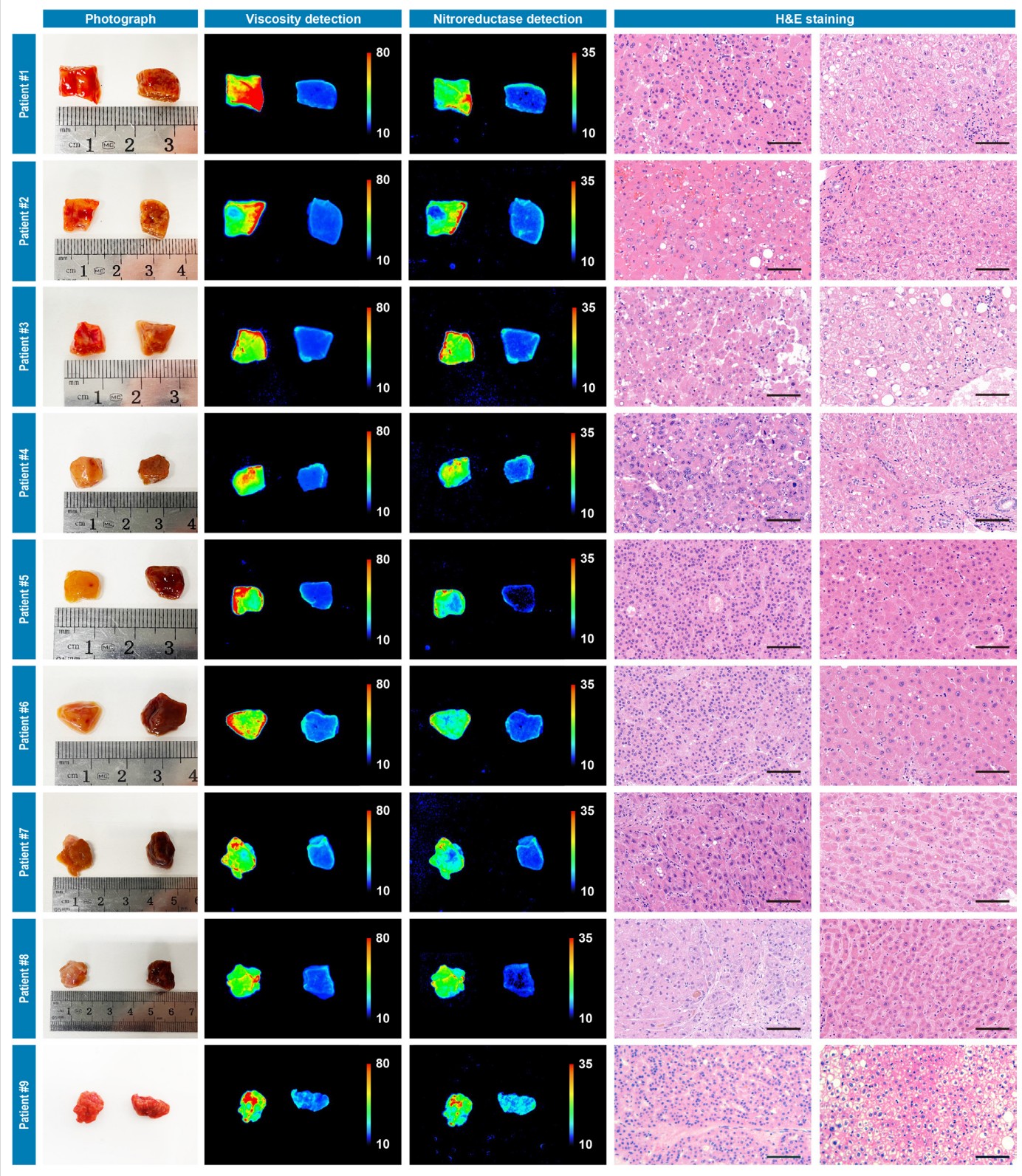

**Figure 4.** Photograph imaging, fluorescence imaging, and hematoxylin and eosin microscopic imaging of nine pairs of cancerous/noncancerous tissues derived from nine individual hepatocellular cancer patients. They were incubated with IBS440 (10 µM) for 20 min and taken imaging. Scale bar: 100 µm.

The online version of this article includes the following figure supplement(s) for figure 4:

**Source data 1.** The original raw data of fluorescence imaging and hematoxylin and eosin staining results of nine pairs of cancerous/noncancerous liver

*Figure 4 continued on next page*

*Figure 4 continued*

tissues.

**Figure supplement 1.** The scatter plot represented the maximal gray-scales signal intensities of nine pairs of liver tissue samples in nitroreductase and viscosity detection channels.

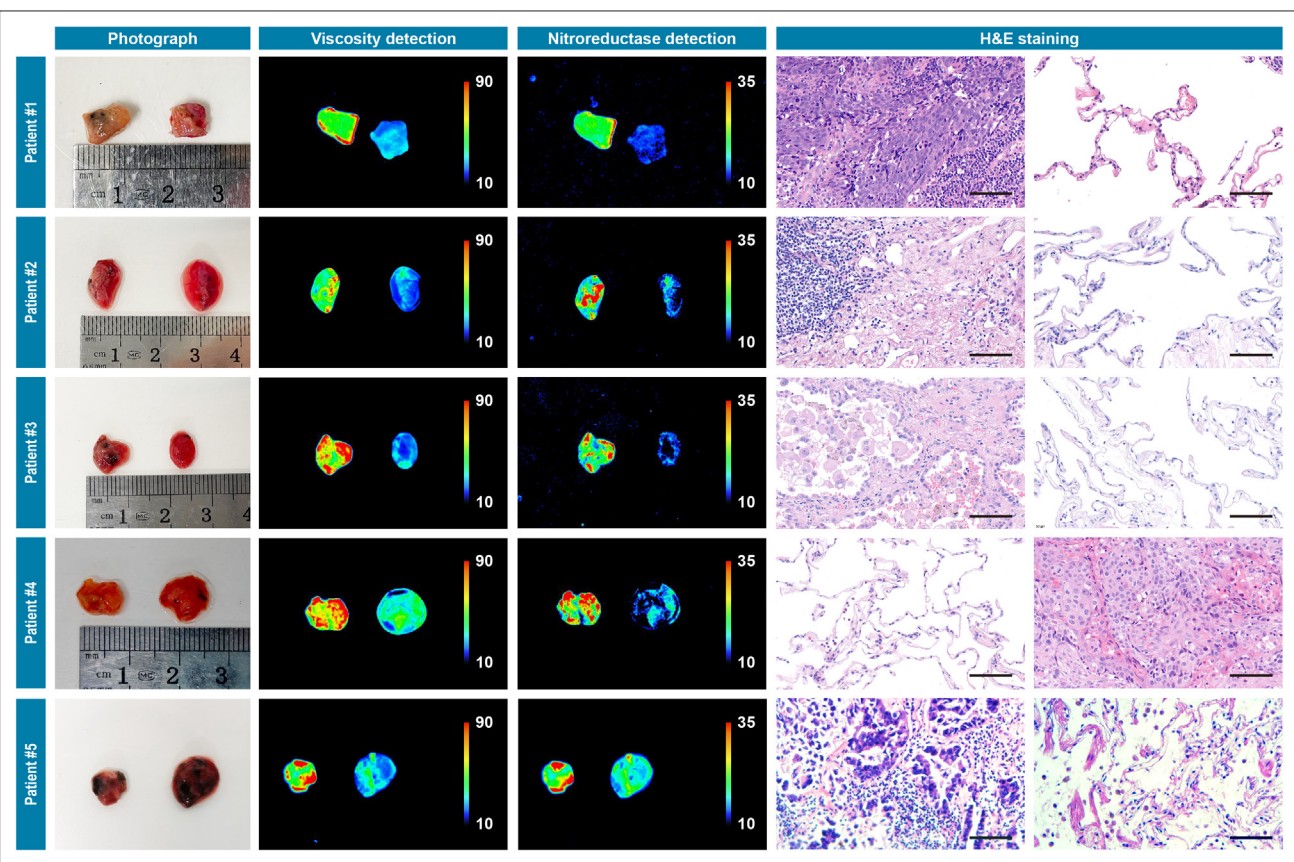

**Figure 5.** Photograph imaging, fluorescence imaging, and hematoxylin and eosin microscopic imaging of five pairs of cancerous/noncancerous tissues derived from five individual lung cancer patients. They were incubated with IBS440 (10 μM) for 20 min and taken imaging. Scale bar: 100 μm.

The online version of this article includes the following figure supplement(s) for figure 5:

**Source data 1.** The original raw data of fluorescence imaging and hematoxylin and eosin staining results of five pairs of cancerous/noncancerous lung tissues.

**Figure supplement 1.** The scatter plot represented the maximal gray-scales signal intensities of five pairs of lung tissue samples in nitroreductase and viscosity detection channels.

**Figure supplement 2.** Photograph imaging, fluorescence imaging, and hematoxylin and eosin microscopic imaging of five pairs of cancerous/noncancerous tissues derived from five individual oral cancer patients.

**Figure supplement 2—source data 1.** The original raw data of fluorescence imaging and hematoxylin and eosin staining results of five pairs of cancerous/noncancerous oral tissues.

**Figure supplement 3.** Photograph imaging, fluorescence imaging, and hematoxylin and eosin microscopic imaging of six pairs of cancerous/noncancerous tissues derived from six individual renal cancer patients.

**Figure supplement 3—source data 1.** The original raw data of fluorescence imaging and hematoxylin and eosin staining results of six pairs of cancerous/noncancerous renal tissues.

**Figure supplement 4.** The scatter plot represented the maximal gray-scales signal intensities of five pairs of oral tissue samples in nitroreductase and viscosity detection channels.

**Figure supplement 5.** The scatter plot represented the maximal gray-scales signal intensities of six pairs of renal tissue samples in nitroreductase and viscosity detection channels.

point, any sample would be located into one of the three divided groups, that is, the definitely positive (++) group showing two high intensities in both x and y axis; the suspiciously positive (+) group showing a high intensity in x or y axis; and negative (-) group showing two low intensities in both x and y axis.

Second, to verify the thresholds, 35 pieces of resection margin tissues derived from one hepatocellular cancer patient and 16 pieces of resection margin tissues derived from one lung cancer patient were used as test datasets, respectively. All samples were incubated with IBS440 and imaged by an in vivo small animal instrument. As shown in *Figures 6A and 7A*, there were differences in fluorescence intensity across samples, as assessed through the viscosity and nitroreductase detection channels. All the samples, prior to pathological examination, were categorized into three groups according to the thresholds (*Figures 4 and 5*): definitely positive (++), strong fluorescence in both channels; suspiciously positive (+), strong fluorescence in one channel; and negative (-), weak fluorescence in both channels (*Figures 6B and 7B*). The hematoxylin and eosin staining of permanent sections were followed to confirm the classification results (*Figures 6C and 7C*, *Figure 6—figure supplement 1* and *Figure 7—figure supplement 1*). Overall, a specificity of 100 % was achieved in screening tumor-free samples from positive ones among the liver and lung tumor samples. There was no test dataset of oral and renal cancer samples due to the small tumor volume and insufficient sampling numbers.

## Discussion

As early as in 1940 s, a study found that the cytoplasm of tumor cells became more viscous than normal cells (*Claus, 1942*), followed by many researches validating its high viscosity (*Yang et al., 2014*; *Ren et al., 2020*; *Song et al., 2021*; *Zhou et al., 2021*). On other hand, it has been well known that hypoxia and nitroreductase induced by hypoxia are recognized as characteristics of tumor cells (*Brown and Wilson, 2004*). Promisingly, the abnormal intracellular changes in viscosity and nitroreductase in tumor cells could be exploited as biomarkers for evaluation of the resected margins of multiple distinct tumors. In this study, we developed a new dual-activatable fluorescent molecular rotor IBS440 enabling visualization of cytoplasmic viscosity and hypoxia of live cells in ex vivo tissues, which, in a rapid and high-throughput manner, distinguishes between tumor and normal tissue in unsectioned surgical specimens.

Fluorescence imaging has been recognized advantageous and clinically exploited to visualize tumor tissues during image-guided open and endoscopic surgery, and pathological assessment, based on the recognition of tumor-specific markers. Abnormal intracellular microenvironment has been described as characteristic of cancer cells for decades, including biophysical and biochemical aspects. Thus, we utilized the abnormal increases in biophysical cytoplasmic viscosity and biochemical endogenous nitroreductase of cancer cells, as two different reactive sites to design our dual-activatable fluorescent rotor. This molecule emits two non-interfering fluorescence signals by activating two different modes, one specifically responses to higher viscosity (emitting red fluorescence) and the other to abundant nitroreductase (emitting green fluorescence). We demonstrated its availability to discriminate tumor-involved margins in surgical specimens of four cancer types including lung, renal, hepatocellular, and oral carcinoma. As a result, the increases in cytoplasmic viscosity and hypoxia of cancer cells were imaged by this dual-activatable fluorescent molecular rotor, indicating its potential feasibility to be used in pre-screening multiple resected tumor margins.

During or after surgery, pathologists rapidly diagnose a few of tissue samples of one patient and immediately convey the results to surgeons while the patient is still in the operating room; and prioritizing specimens for pathological analysis exclusively relies on visual inspection and palpation of the fresh specimen by the surgeons and pathologists, for example color, texture, consistency, nodules (*Hinni et al., 2013*; *Figure 8A*). In practice, it is impractical to implement the thorough frozen-section of the entire specimens and complete microscopic examination on every tissue slice, constrained by limited resource. Sometimes, tumor-involved margins may not be included in the prioritized section blocks, thus causing an inadequate sampling and examination. Therefore, we propose a strategy to pre-screen and prioritize the specimens with integral fluorescence visualization, which might optimize current intraoperative pathological procedures and minimize inadequate sampling. As described in *Figure 8B*, immediately after resection, all sampled specimens are trimmed and simultaneously incubated in a phosphate buffer saline solution of fluorescent molecule. Afterwards, tissues are directly imaged and reviewed on a back-table instrument. As shown in *Figure 8B* (i), a negative result, which

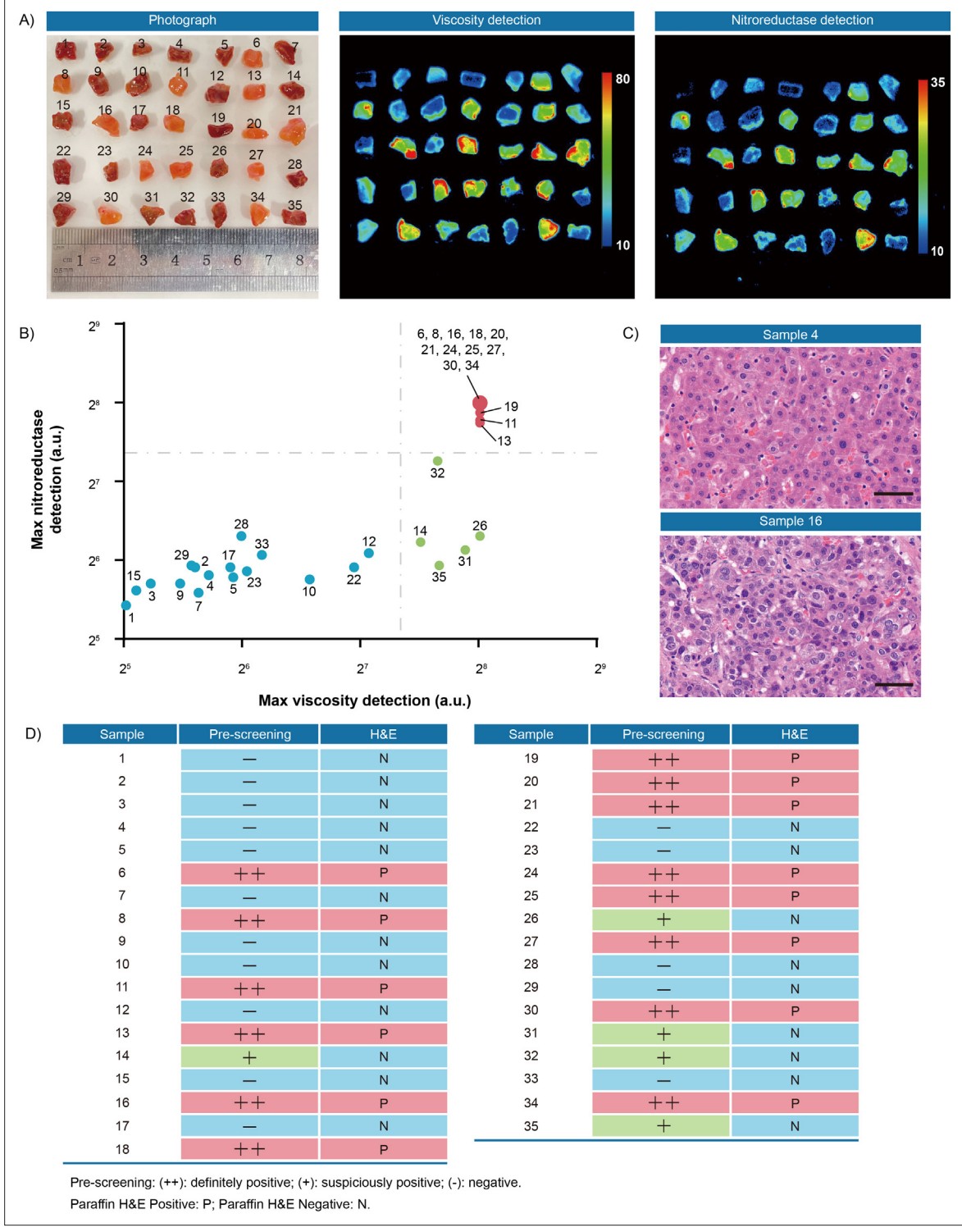

**Figure 6.** Fluorescence pre-screening of 35 pieces of resection margin tissues derived from one hepatocellular cancer patient. (**A**) Photograph imaging and fluorescence imaging of resected tissues. (**B**) Clustering analysis using K-means algorithm on the maximal gray-scales signal intensities of each tissue sample in nitroreductase and viscosity detection channels. (**C**) Hematoxylin and eosin microscopic imaging of the representative liver samples. Scale bar: 50 µm. (**D**) The correlation analysis of liver tissue samples between fluorescence signals and hematoxylin and eosin staining results. Pre-screening: (++): definitely positive; (+): suspiciously positive; (-): negative. Paraffin H&E Positive: P; Paraffin H&E Negative: N.

The online version of this article includes the following figure supplement(s) for figure 6:

**Figure supplement 1.** Hematoxylin and eosin staining results of 35 pieces of resected margin tissues derived from one hepatocellular cancer patient.

**Source data 1.** The original raw data of fluorescence imaging and hematoxylin and eosin staining results of 35 pieces of resection margin tissues.

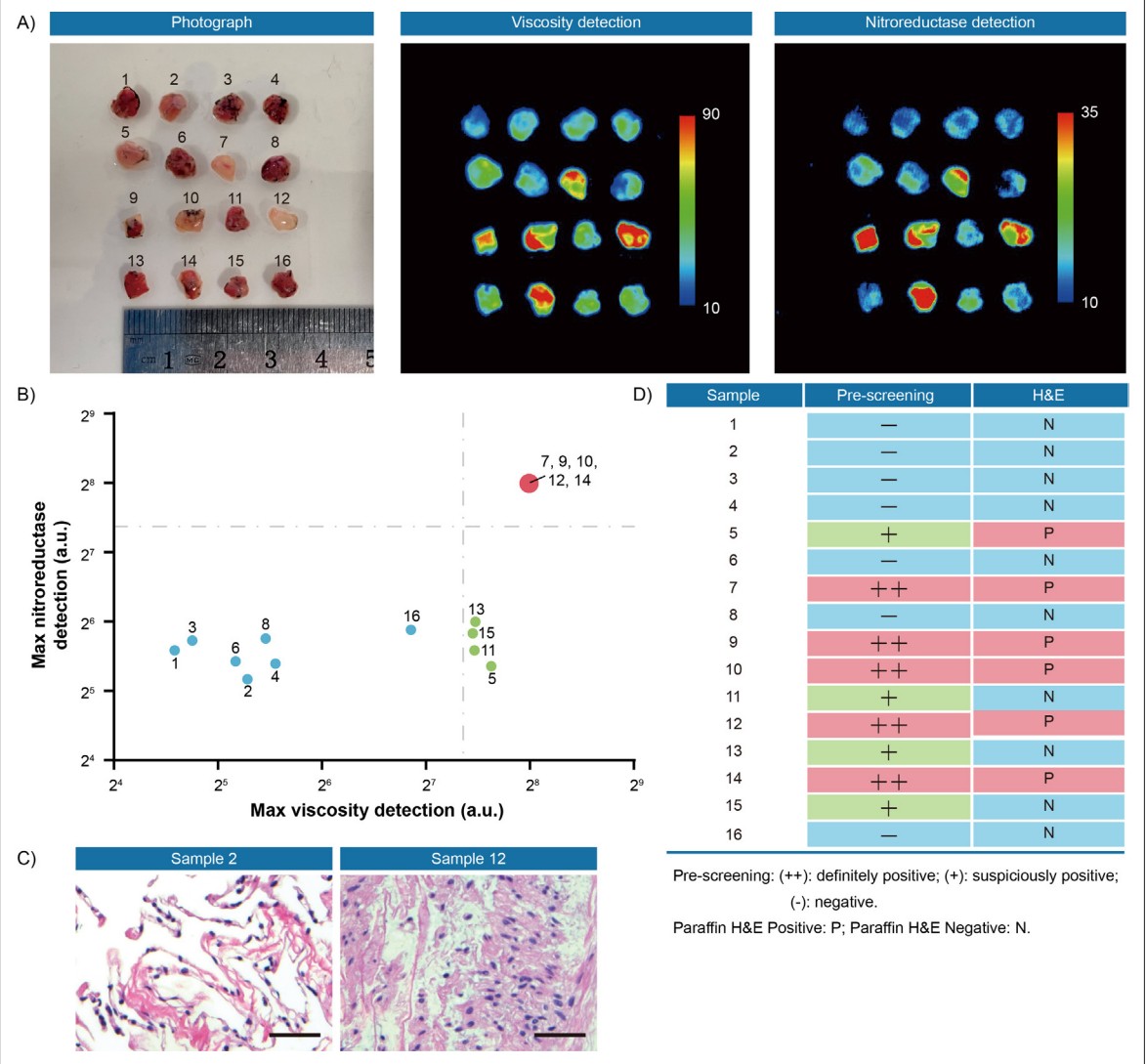

**Figure 7.** Fluorescence pre-screening of 16 pieces of resection margin tissues derived from one lung cancer patient. (**A**) Photograph imaging and fluorescence imaging of resected tissues. (**B**) Clustering analysis using K-means algorithm on the maximal gray-scales signal intensities of each tissue sample in nitroreductase and viscosity detection channels. (**C**) Hematoxylin and eosin microscopic imaging of the representative lung samples. Scale bar: 50 μm. (**D**) The correlation analysis of lung tissue samples between fluorescence signals and hematoxylin and eosin staining results. Pre-screening: (++): definitely positive; (+): suspiciously positive; (-): negative. Paraffin H&E Positive: P; Paraffin H&E Negative: N.

The online version of this article includes the following figure supplement(s) for figure 7:

**Figure supplement 1.** Hematoxylin and eosin staining results of 16 pieces of resected margin tissues derived from one lung cancer patient.

**Source data 1.** The original raw data of fluorescence imaging and hematoxylin and eosin staining results of 16 pieces of resection margin tissues.

is characterized by low fluorescence intensity of all specimens, can be rapidly conveyed to the operating surgeon to close the surgery case. If any specimens display high fluorescence, they are considered as suspiciously positive ones and prioritized for further review (*Figure 8B* (ii)). The fluorescence signal can be visualized by many types of commercial imaging instruments, such as a multifunctional laser scanner as well as a small animal in vivo imaging machine, which can rapidly screen as many as tens and hundreds of tissue fragments at one time. Even an automated high-content fluorescence or confocal fluorescence microscopy can be used upon request, which provides a very high sensitivity and more detailed intracellular information for in-depth assessment. Hopefully, this strategy may reduce the waiting time for intraoperative pathological assessment, thereby shortening operation time, promoting patient recovery, and reducing pathologists' workload.

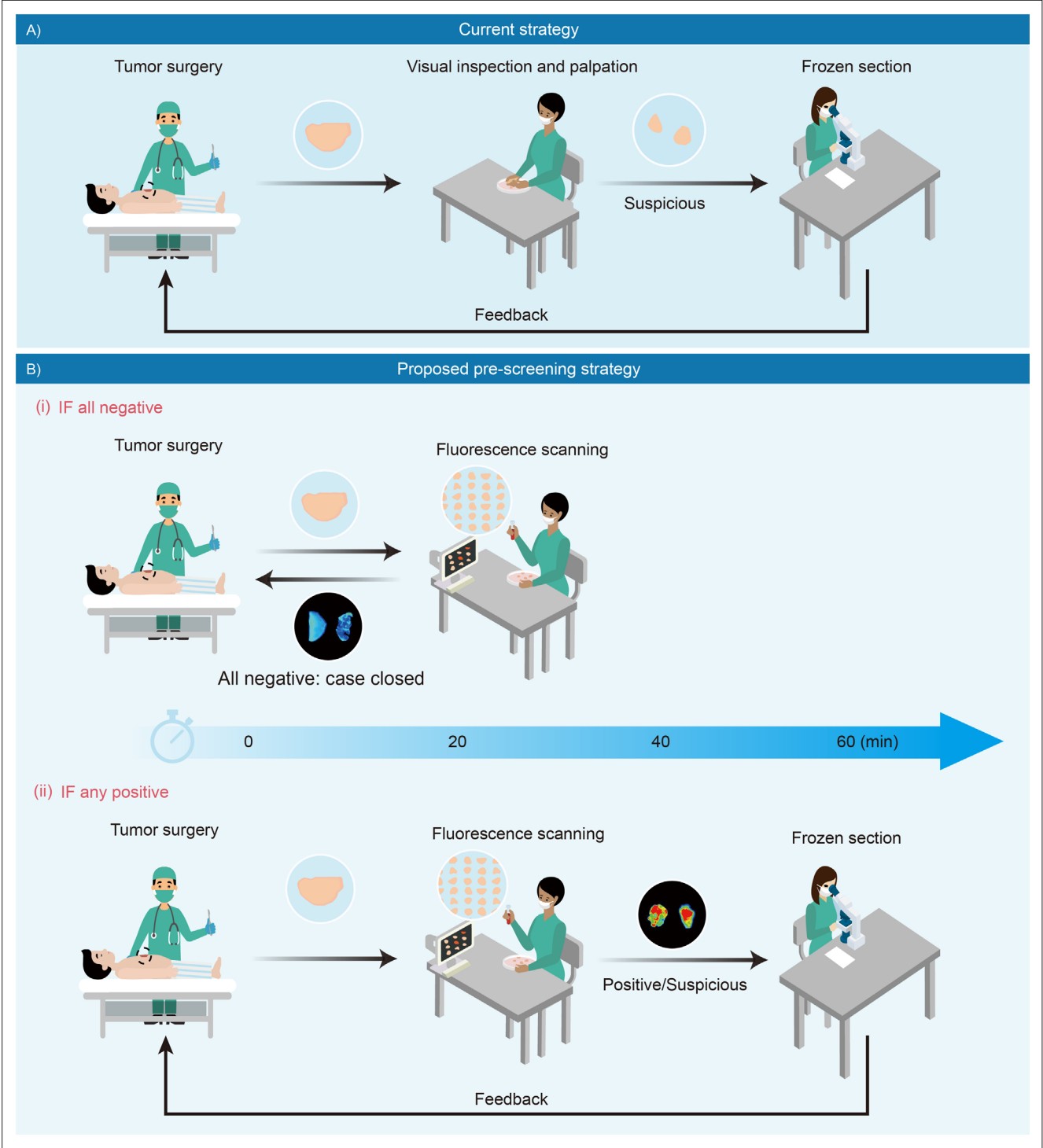

**Figure 8.** Schematic presentation of intraoperative pathological assessment. (**A**) The current strategy used during a tumor operation. Based on surgeon's gross examination, multiple resected tissues are prioritized for frozen-section analysis. Occasionally, an extended resection is indicated with positive feedback to avoid unnecessary re-operations. In principle, frozen section analysis takes minimally 30 min to 1 hr; while in practice, such procedure requires more time to complete due to a large number of samples received in overloaded pathological laboratories. (**B**) Fluorescence-based pre-screening strategy described in this paper. It allows the pre-screening of tumor involvement with high sensitivity by incubating multiple resected ex vivo tissues with chemical fluorescent agent for minutes and then imaged. (**i**) Negative result is rapidly conveyed to the operation room to close the case with 20 min. (**ii**) Positive ones are prioritized for further frozen-section double-checks and reviews.

This study has a few limitations. First, the fluorescence threshold was slightly different among

different types of cancer in this study, this might arise from the differences of organ and cancer types, as well as the sampling numbers. It's worth noting that the fluorescence intensity is highly instrument-specific. Different imaging instruments (cameras) have their different sensitivities. Thus, the threshold is also dependent on the specific instrument and the setting of measuring parameters, calling for further clinical trials on other commercial imaging instruments. Second, the molecule IBS440 is a demonstrative probe, while the performance is still on demand to be improved for potential clinical applications, such as the higher fluorescence sensitivity (*Kuimova et al., 2009*) and redshift toward longer wavelength (*Vahrmeijer et al., 2013*). The future efforts of chemical biologists could be exerted on the invention of more fluorescent molecules with modified fluorescence features.

# Materials and methods

**Key resources table**

| Reagent type (species) or resource | Designation | Source or reference | Identifiers | Additional information |
|---|---|---|---|---|
| Strain, strain background (*Escherichia coli*) | BL21(DE3) | TransGen Biotech | Cat#CD701 | |
| Antibody | Anti-HIF-1A antibody (Rabbit polyclonal) | Sangon Biotech | Cat#D162108 RRID:AB_1674786 | WB (1:500) |
| Antibody | Anti-6 His antibody (Rabbit polyclonal) | Sangon Biotech | Cat#D110002 RRID:AB_10575638 | WB (1:2000) |
| Cell line (*Homo sapiens*) | FaDu cells | Stem Cell Bank, Chinese Academy of Sciences | Cat#TCHu132 | RRID:CVCL_1218 |
| Cell line (*Homo sapiens*) | A549 cells | Stem Cell Bank, Chinese Academy of Sciences | Cat#SCSP-503 | RRID:CVCL_0023 |
| Cell line (*Homo sapiens*) | MHCC97H cells | Zhongshan Hospital, Fudan University | | |
| Cell line (*Mus musculus*) | RAW264.7 cells | Stem Cell Bank, Chinese Academy of Sciences | Cat#TCM13 | RRID:CVCL_0493 |
| Cell line (*Rattus norvegicus*) | A7r5 cells | Stem Cell Bank, Chinese Academy of Sciences | Cat#GNR 7 | RRID:CVCL_0137 |
| Biological sample (*mouse*) | BALB/c nude mouse (4–6 weeks old, female) | School of Pharmacy, Fudan University | http://www.lacsp.fudan.edu.cn/ | |
| Biological sample (*human*) | Lung carcinoma specimens | Zhongshan Hospital, Fudan University | | |
| Biological sample (*human*) | Renal carcinoma specimens | Zhongshan Hospital, Fudan University | | |
| Biological sample (*human*) | Hepatocellular carcinoma specimens | Zhongshan Hospital, Fudan University | | |
| Biological sample (*human*) | Oral carcinoma specimens | Shanghai Ninth People's Hospital | | |
| Chemical compound, drug | 4-picoline | Energy-Chemical | Cat#W3300080250 | CAS: 108-89-4 |
| Chemical compound, drug | potassium tert-butoxide | Energy-Chemical | Cat#E0600050250 | CAS: 865-47-4 |
| Chemical compound, drug | 1-(bromomethyl)–4-nitrobenzene | Energy-Chemical | Cat#W610062 | CAS: 100-11-8 |
| Chemical compound, drug | NaCl | Energy-Chemical | Cat#E010368 | CAS: 7647-14-5 |
| Chemical compound, drug | Nicotinamide Adenine Dinucleotide Phosphate (NADPH) | Energy-Chemical | Cat#10107824001 | CAS: 2646-71-1 |
| Chemical compound, drug | glycerol | Energy-Chemical | Cat#A0404071000 | CAS: 56-81-5 |
| Peptide, recombinant protein | T4 DNA ligase | New England Biolabs | Cat#M0202 | a final concentration of 20 units/μL |
| Peptide, recombinant protein | XhoI | New England Biolabs | Cat#R0146 | a final concentration of 0.4 units/μL |

*Continued on next page*

*Continued*

| Reagent type (species) or resource | Designation | Source or reference | Identifiers | Additional information |
|---|---|---|---|---|
| Peptide, recombinant protein | BamHI-HF | New England Biolabs | Cat#R3136 | a final concentration of 0.4 units/μL |
| Peptide, recombinant protein | Taq DNA polymerase | New England Biolabs | Cat#M0273 | a final concentration of 0.025 units/μL |
| Commercial assay or kit | BCA protein assay kit | Beyotime | Cat#P0012 | |
| Commercial assay or kit | Cell Counting Kit-8 | Beyotime | Cat#C0038 | |
| Commercial assay or kit | gel extraction kit | Tiangen Biotech | Cat#DP209 | |
| Software, algorithm | Chemdraw 20 | PerkinElmer Informatics | https://www.chemdraw.com.cn | RRID:SCR_016768 |
| Software, algorithm | Origin 8.0 | Origin Lab | https://www.originlab.com/ | |
| Software, algorithm | Ilustrator 2021 | Adobe | https://www.adobe.com | |
| Software, algorithm | Image J | Image J | https://imagej.nih.gov/ij/ | RRID:SCR_003070 |
| Other | Hoechst 33,342 | Beyotime | Cat#C1029 | dilution (1:100) |
| Other | Mito-Tracker Green | Beyotime | Cat#C1048 | a final concentration of 1 μM |

Apparatus: The pH of the testing systems was determined by Orion Star A211 pH Meter (USA). The fluorescent spectra were recorded at room temperature on an Edinburgh Photoluminescence Spectrometer FLS1000 (UK). The UV-vis absorption spectra were recorded on a UV-vis spectrometer L6 (China). All cell and tissue slides fluorescence images were acquired by a Leica TCS SP8 STED (Germany). A 405 nm laser was used as the excitation source and the emission wavelength was collected using a blue channel (430–480 nm) for cell nucleus staining dye Hoechst 33342, a 405 nm laser was used as the excitation source and the emission wavelength was collected using a green channel (480–530 nm) for IBS440 recognition of nitroreductase, a 488 nm laser was used as the excitation source and emissions were collected using a green channel (500–550 nm) for Mito-Tracker Green, and 488 nm laser was used as the excitation source and emissions were collected using a red channel (580–630 nm) for IBS440 recognition of viscosity. In vivo animal and organs fluorescence imaging were acquired by an In Vivo Bruker Xtreme (USA).

## Synthesis of IBS224 and IBS440

**Scheme 1.** Synthesis of IBS224 and IBS440.

A mixture of 4-picoline (1.863 g, 20 mmol), potassium tert-butoxide (2.357 g, 21 mmol), and 4-dimethylaminobenzaldehyde (2.984 g, 20 mmol) was introduced into flask of 10 mL anhydrous DMF and heated to 60 °C for 7 h under efficient magnetic stirring. After being cooled at room temperature,

the mixture was added to ice water. Next the mixture was filtered and washed with a mixture of methanol and cold water (v:v = 1:1), then a yellow solid IBS224 was obtained by drying under vacuum.

Yield: 2.908 g (60%). $^1$H NMR (400 MHz, Chloroform-d) δ 8.50 (d, J = 4.9 Hz, 2 H), 7.44 (d, J = 8.7 Hz, 2 H), 7.34 (d, J = 5.9 Hz, 2 H), 7.26 (d, J = 16.3 Hz, 1 H), 6.80 (d, J = 16.2 Hz, 1 H), 6.71 (d, J = 8.8 Hz, 2 H), 3.03 (s, 6 H). $^{13}$C NMR (100 MHz, DMSO-d6) δ 145.51, 143.77, 140.95, 128.66, 124.69, 123.08, 118.70, 115.26, 106.79, 34.91. MS (ES-API): calcd for $C_{15}H_{16}N_2$ m/z [M + H]$^+$ 225.1, found 225.1.

Compound IBS224 (1.122 g, 5 mmol) and 1-(bromomethyl)–4-nitrobenzene (1.082 g, 5 mmol) were dissolved in anhydrous acetonitrile (10 mL). The mixture was refluxed at 85 °C for 12 hr. After the removal of the solvent under reduced pressure, the residue was purified by column chromatography with $CH_2Cl_2$/ Methanol as the eluent from 40:1 to 5:1(v:v) and obtained a red solid product IBS440.

Yield: 1.783 g (81%). $^1$H NMR (400 MHz, DMSO-d6) δ 8.86 (d, J = 6.8 Hz, 2 H), 8.28 (d, J = 8.8 Hz, 2 H), 8.08 (d, J = 6.7 Hz, 2 H), 7.94 (d, J = 16.0 Hz, 1 H), 7.71 (d, J = 8.6 Hz, 2 H), 7.58 (d, J = 8.6 Hz, 2 H), 7.17 (d, J = 16.0 Hz, 1 H), 6.81–6.73 (m, 2 H), 5.80 (s, 2 H), 3.01 (s, 6 H). $^{13}$C NMR (100 MHz, DMSO-d6) δ 154.30, 151.91, 147.58, 143.64, 142.87, 141.78, 130.26, 129.49, 124.00, 122.57, 122.23, 116.80, 111.80, 60.45. calcd for $C_{22}H_{22}N_3O_2$ m/z M$^+$ 360.1, found 360.1.

## Cell culture

MHCC97H, FaDu, A549, RAW264.7 and A7r5 cells were grown in Dulbecco's modified Eagle's medium (DMEM) with 10 % fetal bovine serum (FBS) and 100 U/mL penicillin and 100 U/mL streptomycin. The cells were negative for mycoplasma contamination.

## Cell line derived tumor xenograft mouse models

The BALB/c nude mouse (4–6 weeks old, female) were originally purchased from the SPF experimental Animal Center. All experimental procedures were conducted in accordance with the protocols approved by the Department of Laboratory Animal Science, Fudan University (202012023 S). Three cell lines including MHCC97H cells, FaDu cells and A549 cells were used to form tumor-bearing animal models for the in vivo and ex vivo imaging study. Tumor cells for subcutaneous flank graft were suspended in 100 μL of serum-free media. BALB/c nude mice were injected subcutaneously in the flank with about $1 \times 10^6$ cells of each cell line to form tumor-bearing mouse models, respectively. Tumors with diameters of around 8 mm were formed after 21 days.

## Surgical specimens

Surgical resected tumor specimens, including lung carcinoma, renal carcinoma and hepatocellular carcinoma specimens were collected from Zhongshan Hospital, Fudan University and oral carcinoma specimens were collected from Shanghai Ninth People's Hospital, Shanghai Jiaotong University School of Medicine. Written informed consent and consent to publish were obtained from all patients before participation. All procedures were conducted in accordance with the protocols approved by the Ethics Committee of the Institutes of Biomedical Sciences, Fudan University (2020–014). The tissue samples were immediately used for fluorescence imaging experiments. Nine hepatocellular cancer samples were obtained from nine individual patients (mean age: 49.7 years; range: 39–61 years; 7 males), five lung cancer samples were obtained from five individual patients (mean age: 65.8 years; range: 55–76 years; 3 males), five oral cancer samples were obtained from five individual patients (mean age: 67.8 years; range: 58–78 years; 4 males), and six renal cancer samples were obtained from six individual patients (mean age: 67.8 years; range: 58–78 years; 2 males).The patients' basic information is available in Supplementary file 6.

## Preparation and purification of *E. coli* nitroreductase

Primers were designed according to the nitroreductase gene sequence (NC_000913.3), primer 1 is 5'-GGTGCTCGAGTTACACTTCGGTTAAGGTGA-3', primer 2 is 5'-TCGCGGATCCATGGATATCATTTC TGTCGC-3', and the underscore was the cleavage site of XhoI and BamHI. The nitroreductase gene sequence was amplified in the *Escherichia coli* genome by PCR with primers 1 and 2. The process of PCR began with denaturation at 95 °C for 1 min, 30 cycles of amplification (95 °C for 30 s, 49 °C for 20 s, 72 °C for 30 s), and ended with a single extension step of 72 °C for 3 min. The PCR products were obtained by gel extraction kit and digested by XhoI and BamHI, then cloned to pET-28a. The

correct recombinant pET-28a-nitroreductase was confirmed by sequencing. pET-28a-nitroreductase was transformed into chemically competent *E. coli* BL21 (DE3) cells, an aliquot was plated on SOB agar with kanamycin (50 µg/ml) and grew overnight at 37 °C. A single colony was used for the preculture of which 10 mL was inoculated in 1 L SOB medium with kanamycin (50 µg/ml). The bacterial culture was incubated at 37 °C, 200 rpm until OD 600 of 0.6–0.8 was reached and then changed the temperature to 15 °C for another 2 hr. Then the production of the protein was induced with 0.5 mM IPTG and the culture was incubated at 15 °C, 200 rpm for additional 15 hr. Then *E. coli* cells were harvested by centrifugation (5000 rpm for 15 min at 4 °C). The cell pellet was resuspended in precooled lysis buffer (50 mM Tris-HCl, 300 mM NaCl, 2/10,000 mercaptoethanol, pH 8.0) and lysed by sonication. Cell debris was removed by centrifugation (15,000 rpm for 30 min at 4 °C), the soluble fraction was loaded onto Ni agarose column and the resulting suspention was gently mixed for 2 hr at 4 °C. The column was washed with 3 × 1 bed volumes of pre-cooled lysis buffer and finally the protein was gradient eluted with elution buffer (50 mM Tris-HCl, 150 mM NaCl, 2/10000 mercaptoethano, pH 8.0) containing different imidazole range from 20 mM to 500 mM. Elution buffer was exchanged to storage buffer (50 mM Tris-HCl, 150 mM NaCl, 0.1 mM EDTA, 1 mM DTT, pH 8.0, 10 % glycerol (v/v)). Protein concentration was determined by using BCA protein assay kit.

## Fluorescence and UV-Vis absorption spectroscopy

The solutions for viscosity were prepared by mixing EtOH and glycerol in different proportions. The solutions of IBS440 in different viscosities were prepared by adding the stock solution of IBS440 (1 mM, 50 µL) to EtOH-glycerol mix solution (5 mL) to obtain the final concentration of IBS440 (10 µM). These solutions were sonicated for 30 min to eliminate air bubbles. Then, the fluorescence spectra were recorded at excitation of 500 nm with both the excitation and emission slit widths set at 1 nm. The relationship between the fluorescence emission intensity of IBS440 and the viscosity of the solvent was well expressed by the Forster-Hoffmann equation: $\log(I_f) = c + x \log \eta$, where $I_f$ was the fluorescence intensity, $\eta$ was the viscosity of solution, x and c were constant.

In a reaction buffer solution (50 mM Tris-HCl, 150 mM NaCl, 1 mM DTT, 0.1 mM EDTA, pH 8.0), IBS440 and NADPH at a final working concentration of 10 µM and 500 µM were added. The stock solution of IBS440 was 1 mM in DMSO and NADPH was 5 mM in NaOH (50 mM) solution, respectively. Then, nitroreductase with different concentration was added and the reaction mixture was incubated at 37 °C for 60 min. After that, the mixture was incubated at 80 °C for 5 min to inactivate the enzymes and added with DMSO to increase the solubility of IBS224. Fluorescence spectra were recorded at the excitation wavelength of 400 nm with both the excitation and emission slit widths set at 1 nm. Absorption spectra of IBS440 (10 µM), IBS440 (10 µM)+ NADPH (500 µM)+ NTR (9 µg/mL), IBS224 (10 µM) in reaction buffer, containing 20 % DMSO as a co-solvent were recorded, respectively.

## Cytotoxicity assay

Cell suspension was seeded in 96-well plates and incubated for 24 h (at 37 °C, 5 % CO$_2$). IBS440 and IBS224 of different concentrations (5, 10, 20, 30, 40, 50 µM) were added to the culture plate and incubate for 12 hr and 24 hr. Then the cells were replaced with fresh medium, added with 10 µL CCK8 solution and incubated for 45 min, the absorbance of each plate at 450 nm was recorded.

## Cellular imaging

Three cell lines including MHCC97H cells, FaDu cells and A549 cells were used for cell imaging experiment. To determine the cellular colocalization of IBS440, the cells were treated with IBS440 for 30 min and further stained with 1 µM Mito-Tracker Green for 15 min. The signal of Mito-Tracker Green was collected in a range of 500–550 nm using 488 nm laser. To detect intracellular nitroreductase, cells were grown at 37 °C under normoxic conditions (20 % O$_2$) for 12 hr after being seeded into confocal dishes. Then cells were devided into two groups and incubated under normoxia and hypoxia conditions (2 % O$_2$) for another 12 hr, respectively. After that, the cell medium was then replaced with 5 µM IBS440 in DMEM media and incubated at 37 °C for 20 min. The cells were washed with PBS twice to remove the free IBS440. All cell images were obtained with excitation at 405 nm and emission in a range of 480–530 nm. Hoechst 33,342 (blue channel) was also used to stain the cell nuclei to improve image visualization.

To detect cellular viscosity, the cancerous cells (MHCC97H, A549 and FaDu) and noncancerous cells (RAW264.7, A7r5) were stained with IBS440 (5 µM) for 20 min, then the cells were washed with PBS twice to remove the free IBS440 and taken confocal fluorescence image. The viscosity detection channel was obtained with excitation at 488 nm and emission in a range of 580–630 nm. Hoechst 33,342 fluorescence was collected with excitation at 405 nm and emission at 430–480 nm.

To test the uptake of Mito-Tracker Green in different cells, the cancerous cells (MHCC97H, A549 and FaDu) and noncancerous cells (RAW264.7, A7r5) were incubated with a commercial lipophilic mitochondrial dye Mito-Tracker Green (1 µM) for 20 min, then the cells were washed with PBS twice and taken image. The fluorescence signal of Mito-Tracker Green was collected in an emission range of 500–550 nm using 488 nm laser.

## In vivo and ex vivo fluorescence imaging

Briefly, 100 µL of 10 µM IBS440 solution was injected in situ of tumor. Before imaging, the tumor-bearing mice were anesthetized with isoflurane. Fluorescence imaging was acquired with excitation of 500 nm and collected with 600 nm emission filter for in vivo viscosity detection. After about 20 min, fluorescence imaging was acquired with excitation of 410 nm and collected with 560 nm emission filter for nitroreductase detection.

After in vivo imaging, the mouse was euthanized, major organs including heart, lung, liver, spleen, and kidney were excised and incubated 20 min with IBS440 (10 µM), washed with PBS twice before ex vivo imaging acquisition.

The surgical specimens were trimmed and incubated with IBS440 solution (10 µM) for 20 min, washed with PBS twice and then taken for fluorescence imaging, the incubation condition was the same for different cancer specimens.

Imaging of resected tumor specimens was performed using a fluorescence imaging device (In Vivo Xtreme, Bruker). Exposure type: Standard, Exposure time: 2 s, fStop: 1.4. Using an image processing program (Image J), max fluorescence intensity (gray-scales of images) of every specimen was recorded.

## Histological staining

The mouse tumor, major mouse organs including heart, liver, spleen, lung and kidney, and surgical specimens were sectioned into 2 mm tissue sections and fixed with 10 % paraformaldehyde for 24 hr. Subsequently, the tissue sections were embedded in paraffin and representative 5 µm sections were cut for routine hematoxylin and eosin staining and evaluation.

## Immunoblotting analysis

Cell suspensions (1 × 10⁶) were counted and cultured in six-well plates under normoxic (20 % $O_2$) or hypoxia (2 % $O_2$) conditions. Being cultured for 12 hr or 24 hr, cells were digested and washed with PBS, then resuspended in 1× SDS protein loading buffer, denatured at 100 °C for 10 min and centrifuged for 5 min to remove the cell fragments. The samples were separated by 12 % SDS-PAGE gel and transferred to PVDF membranes. The membranes were blocked with 5 % skimmed milk in PBS-T for 2 hr, then incubated with primary antibodies at 4 °C overnight. HRP-conjugated secondary antibody was used for detection.

## Acknowledgements

This work was supported by grants from the National Key R&D Program of China (2018YFC1005002), the National Natural Science Foundation of China (82070482, 81772007, 21734003, and 51927805), the Shanghai Municipal Science and Technology Major Project (Grant No. 2017SHZDZX01), the Science and Technology Commission of Shanghai Municipality (17JC1400200), and the Shanghai Municipal Education Commission (Innovation Program 2017-01-07-00-07-E00027). We thank Yalin Huang and Jin Li from Institutes of Biomedical Sciences, Fudan University for their assistance at confocal fluorescence imaging.

## Additional information

### Funding

| Funder | Grant reference number | Author |
|---|---|---|
| National Key Research and Development Program of China | 2018YFC1005002 | Weijia Zhang |
| National Natural Science Foundation of China | 82070482 81772007 21734003 51927805 | Weijia Zhang |
| Shanghai Municipal Education Commission | 2017-01-07-00-07-E00027 | Weijia Zhang |
| Shanghai Municipal Science and Technology Major Project | 2017SHZDZX01 | Weijia Zhang |

The funders had no role in study design, data collection and interpretation, or the decision to submit the work for publication.

### Author contributions

Hui Huang, Conceptualization, Investigation, Methodology, Writing - original draft, Writing - review and editing; Youpei Lin, Wenrui Ma, Investigation, Methodology, Resources, Writing - review and editing; Jiannan Liu, Jing Han, Xiaoyi Hu, Mieradilijiang Abudupataer, Investigation, Methodology, Resources; Meilin Tang, Methodology, Resources; Shiqiang Yan, Investigation, Software, Visualization; Chenping Zhang, Qiang Gao, Investigation, Resources; Weijia Zhang, Conceptualization, Funding acquisition, Investigation, Project administration, Supervision, Writing - review and editing

### Author ORCIDs

Hui Huang (ID) http://orcid.org/0000-0002-6817-5611
Mieradilijiang Abudupataer (ID) http://orcid.org/0000-0002-5421-9820
Weijia Zhang (ID) http://orcid.org/0000-0001-6928-0416

### Ethics

Written informed consent and consent to publish were obtained from all patients before participation. All procedures were conducted in accordance with the protocols approved by the Ethics Committee of the Institutes of Biomedical Sciences, Fudan University (2020-014).
Animal model procedures were conducted in accordance with the protocols approved by the Department of Laboratory Animal Science, Fudan University (202012023S).

### Decision letter and Author response

Decision letter https://doi.org/10.7554/eLife.70471.sa1
Author response https://doi.org/10.7554/eLife.70471.sa2

## Additional files

### Supplementary files
• Transparent reporting form
• Supplementary file 1. Clinical Characteristics of the patients.
• Source data 1. Chemical structure characterization.

### Data availability

All data generated or analysed during this study are included in the manuscript and supplementary files. Source data files have been provided.

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

## Appendix 1

### Chemical structures were characterized by $^1$H NMR, $^{13}$C NMR and MS spectra

$^1$H NMR and $^{13}$C NMR spectra were recorded on a Bruker Avance-III 400 instrument (400 MHz for $^1$H NMR and 100 MHz for $^{13}$C NMR spectroscopy). Chemical shifts were reported in ppm (δ) and referenced with respect to residual solvent (DMSO-d6 = 2.50 ppm) for $^1$H-NMR and (DMSO-d6 = 40.0 ppm) for $^{13}$C NMR. $^1$H NMR peaks were referred to as singlet (s), doublet (d), triplet (t) or multiplet (m). Coupling constants (J) were reported in hertz. Mass spectra (MS) were recorded on Agilent LC/MS 6120B.

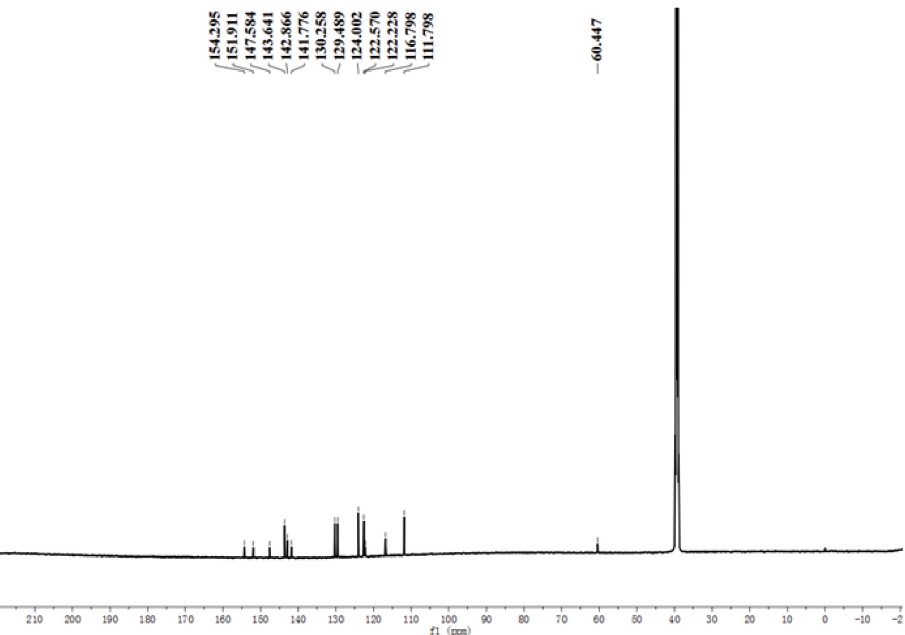

**Appendix 1—figure 1.** $^{13}$C-NMR (DMSO-d6) spectrum of IBS440.

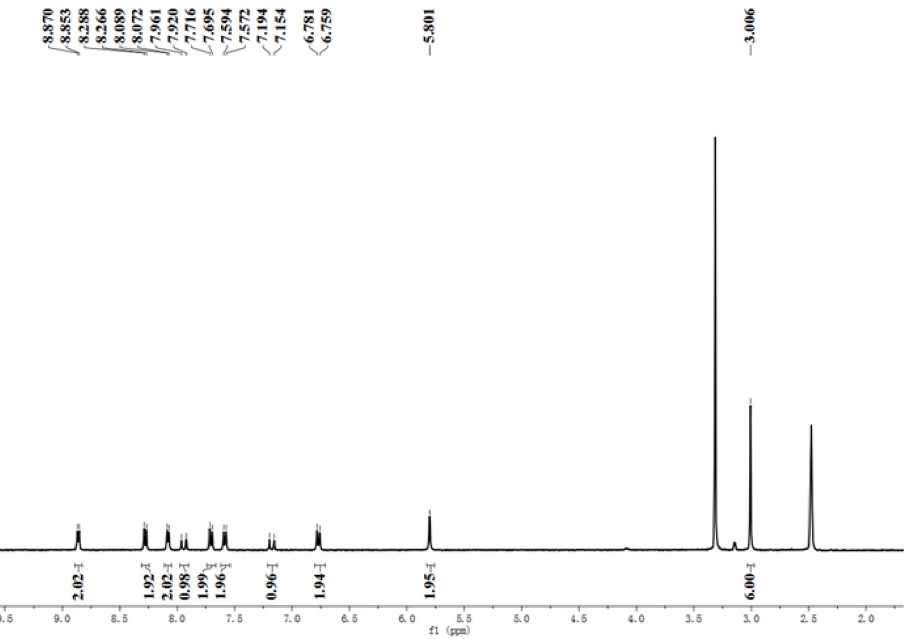

**Appendix 1—figure 2.** $^1$H-NMR (DMSO-d6) spectrum of IBS440.

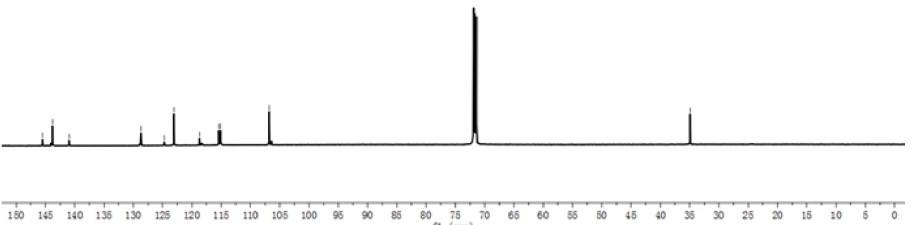

**Appendix 1—figure 3.** $^{13}$C-NMR (DMSO-d6) spectrum of IBS224.

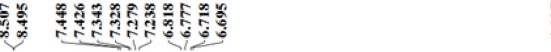

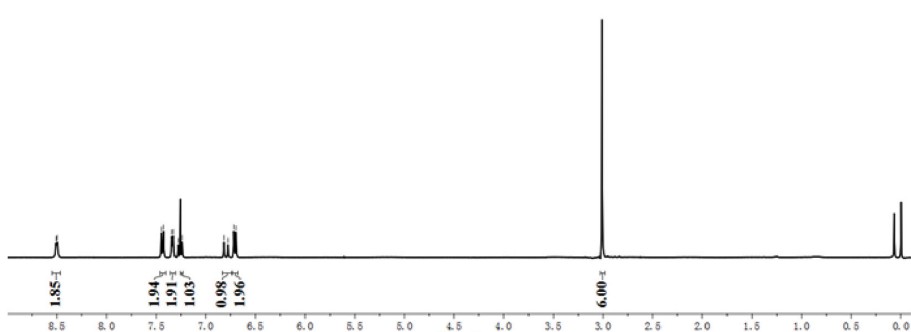

**Appendix 1—figure 4.** $^{1}$H-NMR (CDCl$_3$) spectrum of IBS224.

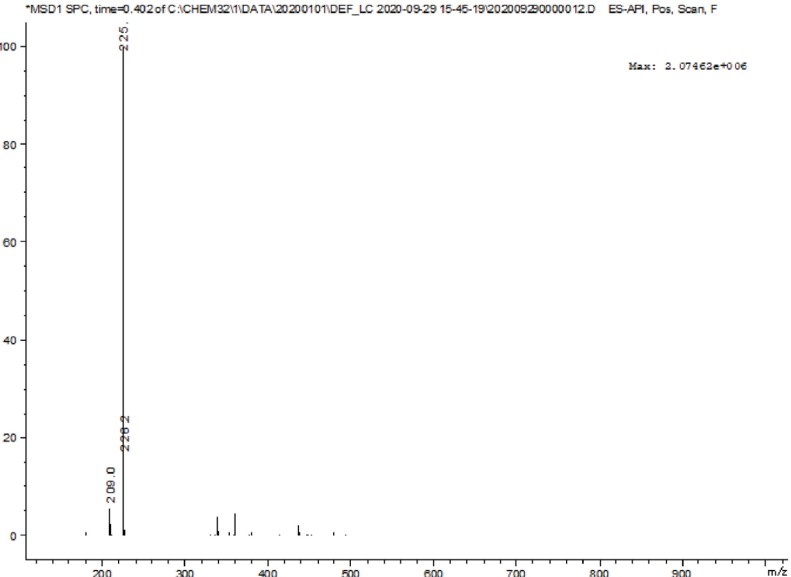

**Appendix 1—figure 5.** MS spectrum of IBS224.

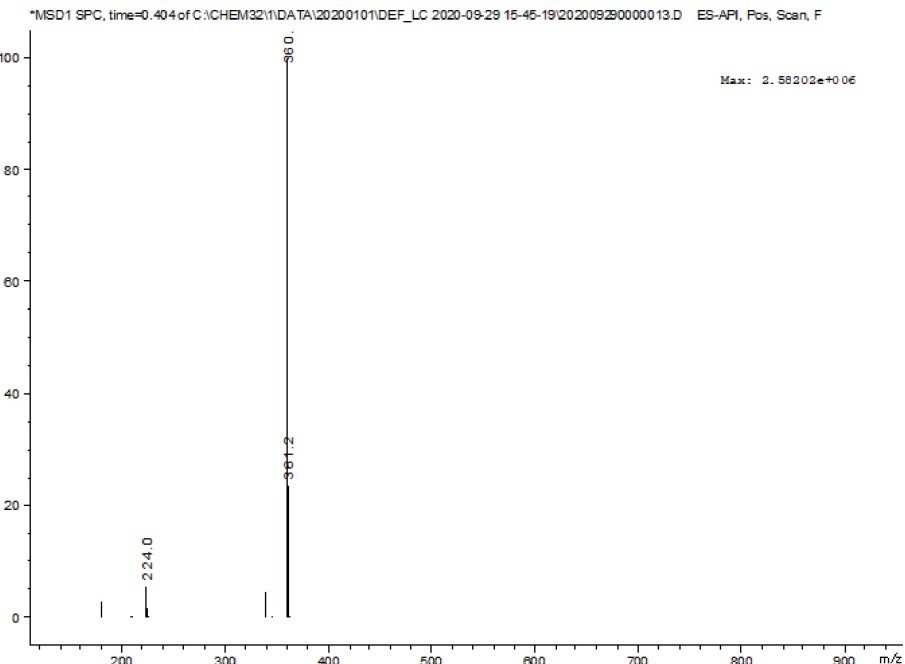

**Appendix 1—figure 6.** MS spectrum of IBS440.

