## [Decision Letter]

**Acceptance summary:**

One novel dual-activatable fluorescent molecular rotor was introduced to optimize currently applied intra-operation frozen tissue section pathological analysis. It takes advantage of the biophysical and biochemical alterations in tumor cells to visualize tumor margin within the whole resected tissue directly and achieves considerable diagnostic efficacy, which shows promising future of translation and application.

**Decision letter after peer review:**

Thank you for submitting your article "A Pre-Screening Strategy to Assess Resected Tumor Margins by Imaging Cytoplasmic Viscosity and Hypoxia" for consideration by *eLife*. Your article has been reviewed by 2 peer reviewers, one of whom is a member of our Board of Reviewing Editors, and the evaluation has been overseen by Mone Zaidi as the Senior Editor. The following individual involved in review of your submission has agreed to reveal their identity: Yi Xiao (Reviewer #3).

Essential revisions:

1) The BLI threshold determined across different cancer types is required for detailed research and analysis, so as the consistency between the same cancer type.

2) It is necessary to demonstrate briefly the cause for difference in fluorescent intensity between normal and malignant cells, and exclude other factors contributing to this difference.

3) More details are required regarding the processing procedure, as well as the discussion on their clinical application efficacy compared with pathological assessment.

*Reviewer #1 (Recommendations for the authors):*

This paper describes in detail the synthesis process and cancer detection principle of IBS440, which has good clinical translation value. The data are well presented, but the manuscript needs more details of clinical application, more clinical cases added for the same type of cancer, and standardization of BLI thresholds for the same type of cancer to ensure the stability of experimental results, and if not, the reasons should be explained.

Considering that pathological diagnosis is the gold standard for cancer diagnosis, relying solely on fluorescence-negative results cannot replace traditional pathological diagnosis, the authors should discuss the clinical application process of IBS440 in detail.

1. The experimental methodology should clearly clarify the incubation time and conditions, the catalog number and manufacturer of the filming equipment, including cytology, in vivo, and ex vivo tumors, as well as the catalog number and manufacturer of the reagents, and the source of the cells. For example, how long the incubation was for image acquisition, and whether there were steps such as PBS washing before image acquisition. Are the incubation conditions the same for different cancers?

2. Are the lung or liver cancer samples in the manuscript from the same patient? Did the authors discuss the differences in sensitivity across pathology types?

3. The BLI thresholds for different types of cancer samples were different when the results were presented in Figure4, how were the BLI thresholds for different samples determined?

4. The BLI thresholds of lung and liver cancer samples in Figure4 are different from the BLI thresholds of lung and liver cancer samples in Figure5 and 6 later, should the BLI thresholds of the same type of cancer be consistent?

5. Pathological diagnosis is the gold standard for cancer diagnosis, relying solely on fluorescence negative results cannot replace the traditional pathological diagnosis, and the Discussion section should discuss the clinical application process of IBS440 in detail.

*Reviewer #3 (Recommendations for the authors):*

It is a good idea to use a single fluorescent probe with dual activatable emissions to differentiate tumors from normal tissues, which provides a strategy for rapid pre-screening in operating room. In this manuscript, the probe IBS400 in tumors demonstrates apparently stronger fluorescence signals of two spectral regions, than those in normal tissues, which qualifies it as a practical probe.

---

## [Author Response]

Essential revisions:1) The BLI threshold determined across different cancer types is required for detailed research and analysis, so as the consistency between the same cancer type.

To address editors’ and reviewers’ concerns, we reconsidered the threshold issue and analyzed more sample tissues derived from 21 individual patients. All samples were divided into training datasets and test datasets for determining the thresholds.

Taking hepatocellular cancer as an example, we used 9 pairs of cancerous/noncancerous tissues derived from 9 individual patients as a training dataset. As shown in Figure 4, the centroid of the signal intensities (gray-scales of images), computed by K-means clustering algorithm, were determined as the threshold. According to the threshold point, any sample would be located into 1 of the 3 divided groups:

i. The definitely positive (++) group showing two HIGH intensities in both x AND y axis;

ii. The suspiciously positive (+) group showing a high intensity in x OR y axis;

iii. The negative (-) group showing two LOW intensities in both x AND y axis.

In the test dataset, 35 pieces of random resection margin tissues derived from one patient were categorized in three groups respectively, as shown in Figure 6.

The same validation of training-and-test model was done on the lung cancer samples (Figure 5 and Figure 7). In the cases of oral and renal cancers, due to the small tumor volume and insufficient sampling numbers, we only included training datasets but not test datasets (Supplementary file 5a and 5b). The calculated thresholds of fluorescence intensity were slightly different (<1% in grayscale, as show in Figure 4—figure supplement 1, Figure 5—figure supplement 1, Supplementary file 5c and Supplementary file 5d) across different cancer types. This might arise from the differences of organ and cancer types, as well as the sampling numbers.

It's worth noting that the fluorescence intensity is highly instrument-specific. Different imaging instruments (cameras) have their different sensitivities. Thus, the threshold is also dependent on the specific instrument and the setting of measuring parameters. The above statement is added in the discussion of the revised manuscript.

In the revised manuscript:

1. We added new sample analyses and re-arranged Figure 4, Figure 5. The original Figure 5 and Figure 6 was re-arranged as Figure 6 and Figure 7.

2. We revised the “Ex vivo imaging of patients’ resected tumor specimens” part in the Results section*,* Page 12, Line 325-360:

“The primary aim of our study was to provide assessments of tumor-involved margins in surgical specimens, thus we measured four types of patients’ resected specimens including hepatocellular cancer, lung cancer, oral cancer and renal cancer. […] There was no test dataset of oral and renal cancer samples due to the small tumor volume and insufficient sampling numbers.”

We discussed the limitation in the Discussion section, Page 18, Line 486-496:

“This study has a few limitations. First, the fluorescence threshold was slightly different among different types of cancer in this study, this might arise from the differences of organ and cancer types, as well as the sampling numbers. […] The future efforts of chemical biologists could be exerted on the invention of more fluorescent molecules with modified fluorescence features.”

2) It is necessary to demonstrate briefly the cause for difference in fluorescent intensity between normal and malignant cells, and exclude other factors contributing to this difference.

To clarify it, we performed more cell and tissue testing and compared the fluorescence of cancerous *vs.* noncancerous cells and tumor *vs.* nontumor tissues, incubated with IBS440, or, a commercial mitochondrial dye Mito-Tracker Green, which is the same kind of lipophilic molecules to target mitochondria.

(1) To demonstrate the different fluorescence intensity between cancerous and noncancerous cells in viscosity detection channel, we performed the fluorescence imaging with IBS440. We added a new paragraph “Fluorescence imaging of viscosity in live cells” in Results section, Page 8, Line 241-252:

“We further studied the fluorescence of IBS440 in viscosity detection channel of cancerous and noncancerous cells. MHCC97H, A549, FaDu, RAW264.7 and A7r5 cells were chosen for fluorescence imaging by confocal microscopy. As shown in Figure 2—figure supplement 6, the cancerous cells (MHCC97H, A549 and FaDu) and noncancerous cells (RAW264.7, A7r5) were stained with IBS440, the fluorescence intensity of the cancerous cells was stronger than that of the noncancerous cells. Specifically, the normal tissues of mice (including heart, liver, spleen, lung, kidney, muscle, fat and brain) and MHCC97H tumor tissue were treated with IBS440, respectively. In comparison with the dim fluorescence signals from normal tissues, the tumor tissue exhibited significantly enhanced fluorescence (Figure 2—figure supplement 7), implying the increased viscosity of tumor tissue.”

(2) To exclude other factors that might affect the fluorescence intensity, such as cellular uptake capacity, we performed the imaging experiment of cells and tissues with a fluorescent dye Mito-Tracker Green. The commercial lipophilic mitochondrial dye Mito-Tracker Green and our IBS440 have the same mechanism of cellular uptake and mitochondrial targeting. We added a new paragraph “Fluorescence imaging of Mito-Tracker Green in live cells and tissues” in Results section, Page 7, Line 216-228:

“Fluorescent small molecules are effectively used in imaging fields because of simple synthesis, high sensitivity and noninvasiveness. Because of their lipophilicity and small molecular weight, the way that they enter cells mainly depends on passive diffusion, which is one of the most common and important transportation methods in lipophilic molecule transportation (Bressloff and Newby, 2013). As shown in Figure 2—figure supplement 2, the cancerous cells (MHCC97H, A549 and FaDu) and noncancerous cells (RAW264.7, A7r5) were incubated with a commercial lipophilic mitochondrial dye Mito-Tracker Green, the result shows the fluorescence intensity in these cells have no significant difference. As shown in Figure 2—figure supplement 3, hepatocellular cancer and normal tissues were incubated with Mito-Tracker Green, and there was also no significant difference in fluorescence intensity. These results showed that the uptake of lipophilic small molecules by cancerous cells and noncancerous cells is at the very similar level.”

(3) We added some other pioneering works on viscosity probes to demonstrate the higher intracellular viscosity in tumor cells than normal cells in Discussion section, Page 17, Line 434-436:

“As early as in 1940s, a study found that the cytoplasm of tumor cells became more viscous than normal cells (P.E.Claus, 1942), followed by many researches validating its high viscosity (Yang et al., 2014, Ren et al., 2020, Song et al., 2021, Zhou et al., 2021).”

Reference:

Ren M, Xu Q, Wang S, Liu L, and Kong F. 2020. A biotin-guided fluorescent probe for dual-mode imaging of viscosity in cancerous cells and tumor tissues. *Chemical communications*, 56: 13351–13354. DOI: https://doi.org/10.1039/d0cc05039c, PMID: 33030195

Song Y, Zhang H, Wang X, Geng X, Sun Y, Liu J, and Li Z. 2021. One Stone, Three Birds: pH Triggered Transformation of Aminopyronine and Iminopyronine Based Lysosome Targeting Viscosity Probe for Cancer Visualization. *Analytical chemistry*, 93: 1786–1791. https://doi.org/10.1021/acs.analchem.0c04644, PMID: 33373187

Zhou Y, Liu Z, Qiao G, Tang B, and LI P. 2021. Visualization of endoplasmic reticulum viscosity in the liver of mice with nonalcoholic fatty liver disease by a near-infrared fluorescence probe. *Chinese Chemical Letters*, DOI: https://doi.org/10.1016/j.cclet.2021.04.035

3) More details are required regarding the processing procedure, as well as the discussion on their clinical application efficacy compared with pathological assessment.

(1) We added the detailed information of reagents, cells and apparatus in Materials and methods section, Key Resources Table.

(2) We added the incubation conditions and more detailed instrument parameters for image acquisition procedure in Materials and methods section, Page 28, Line 648-658:

“To detect cellular viscosity, the cancerous cells (MHCC97H, A549 and FaDu) and noncancerous cells (RAW264.7, A7r5) were stained with IBS440 (5 μM) for 20 min, then the cells were washed with PBS twice to remove the free IBS440 and taken confocal fluorescence image. The viscosity detection channel was obtained with excitation at 488 nm and emission in a range of 580-630 nm. Hoechst 33342 fluorescence was collected with excitation at 405 nm and emission at 430-480 nm.

To test the uptake of Mito-Tracker Green in different cells, the cancerous cells (MHCC97H, A549 and FaDu) and noncancerous cells (RAW264.7, A7r5) were incubated with a commercial lipophilic mitochondrial dye Mito-Tracker Green (1 μM) for 20 min, then the cells were washed with PBS twice and taken image. The fluorescence signal of Mito-Tracker Green was collected in an emission range of 500-550 nm using 488 nm laser.”

Page 28, Line 665-674:

“After in vivo imaging, the mouse was euthanized, major organs including heart, lung, liver, spleen and kidney were excised and incubated 20 min with IBS440 (10 μM), washed with PBS twice before ex vivo imaging acquisition.

The surgical specimens were trimmed and incubated with IBS440 solution (10 μM) for 20 min, washed with PBS twice and then taken for fluorescence imaging, the incubation condition was the same for different cancer specimens.

Imaging of resected tumor specimens was performed using a fluorescence imaging device (in vivo Xtreme, Bruker). Exposure type: Standard, Exposure time: 2 s, fStop: 1.4. Using an image processing program (Image J), max fluorescence intensity (gray-scales of images) of every specimen was recorded.”

(3) The original Figure 7 was re-arranged as Figure 8.

(4) We discussed the clinical application potential and current limitations of this work in Discussion section*:*

“During or after surgery, pathologists rapidly diagnose a few of tissue samples of one patient and immediately convey the results to surgeons while the patient is still in the operating room; and prioritizing specimens for pathological analysis exclusively relies on visual inspection and palpation of the fresh specimen by the surgeons and pathologists, e.g. color, texture, consistency, nodules (Hinni et al., 2013) (Figure 8A). […] The future efforts of chemical biologists could be exerted on the invention of more fluorescent molecules with modified fluorescence features.”